# IKBKB reduces huntingtin aggregation by phosphorylating serine 13 via a non-canonical IKK pathway

Cristina Cariulo[1] , Paola Martufi[1], Margherita Verani[1], Leticia Toledo-Sherman[2,3], Ramee Lee[4], Celia Dominguez[4], Lara Petricca[1,*], Andrea Caricasole[1,*]

N-terminal phosphorylation at residues T3 and S13 is believed to have important beneficial implications for the biological and pathological properties of mutant huntingtin, where inhibitor of nuclear factor kappa B kinase subunit beta (IKBKB) was identified as a candidate regulator of huntingtin N-terminal phosphorylation. The paucity of mechanistic information on IKK pathways, together with the lack of sensitive methods to quantify endogenous huntingtin phosphorylation, prevented detailed study of the role of IKBKB in Huntington's disease. Using novel ultrasensitive assays, we demonstrate that IKBKB can regulate endogenous S13 huntingtin phosphorylation in a manner, dependent on its kinase activity and known regulators. We found that the ability of IKBKB to phosphorylate endogenous huntingtin S13 is mediated through a non-canonical interferon regulatory factor3–mediated IKK pathway, distinct from the established involvement of IKBKB in mutant huntingtin's pathological mechanisms mediated via the canonical pathway. Furthermore, increased huntingtin S13 phosphorylation by IKBKB resulted in decreased aggregation of mutant huntingtin in cells, again dependent on its kinase activity. These findings point to a non-canonical IKK pathway linking S13 huntingtin phosphorylation to the pathological properties of mutant huntingtin aggregation, thought to be significant to Huntington's disease.

## Introduction

In Huntington's disease (HD), an expansion of a CAG repeat within exon 1 of the *huntingtin* (*HTT*) gene, which produces a huntingtin (HTT) protein with an expanded polyglutamine (polyQ) repeat, leads to a progressive and fatal neurodegenerative pathology (Ross & Tabrizi, 2011; Bates et al, 2015; McColgan & Tabrizi, 2018). Mutant HTT fragments comprising the polyQ repeat, produced by alternative splicing and/or proteolytic cleavage, are widely believed to contribute significantly to HD through the propensity of such fragments to misfold and form aggregates (Mangiarini et al, 1996; Landles et al, 2010; Sathasivam et al, 2013; Bates et al, 2015).

HD onset and progression are known to be influenced by a number of modifiers (Flower et al, 2019; Genetic Modifiers of Huntington's Disease Consortium, 2019; Goold et al, 2019; Holmans et al, 2017; Lee et al, 2017), some of which may influence pathology by facilitating CAG-repeat expansion in somatic cells. Other evidence indicates that disease modification may be attainable through mechanisms directly affecting mutant HTT. Hence, protein sequences flanking the polyQ repeat—such as the first 17 residues of the HTT protein (N17 domain) and the proline-rich domain at the N- and C-termini of the polyQ repeat, respectively—has been shown to profoundly alter the biological and biophysical properties of these mutant HTT fragments in isolated proteins, cells and of the full length mutant HTT protein in vivo (Saudou & Humbert, 2016; Chatterjee et al, 2021). Because the N17 domain is the target of several post-translational modifications (PTMs; [Ehrnhoefer et al, 2011; Saudou & Humbert, 2016]), significant attention has focussed on residues associated with PTMs that may be modulated pharmacologically by targeting relevant enzymes, with the aim of altering the pathological properties of mutant HTT. Indeed, studies with phosphor-mimetic or phosphor-abrogative substitutions of residues T3 and S13/S16 have revealed a key role for these residues in regulating mutant HTT properties in isolated proteins (Kelley et al, 2009; Mishra et al, 2012; Crick et al, 2013), cells (Aiken et al, 2009; Thompson et al, 2009; Maiuri et al, 2013; Zheng et al, 2013; Branco-Santos et al, 2017), and mice, where phosphor-mimetic mutants invariably mitigate the effects of polyQ expansion. More recently, the availability of HTT proteins bearing bona-fide phosphorylation at these residues has confirmed the key role for T3 and S13/S16 in modulating mutant HTT phenotypes in vitro (Cariulo et al, 2017; Chiki et al, 2017; DeGuire et al, 2018). Parallel advances in developing assays that can specifically detect and quantify HTT phosphorylation have provided the required tools to study these

---

[1]Neuroscience Unit, Translational and Discovery Research Department, IRBM S.p.A., Rome, Italy  [2]Rainwatercf.org Tau Consortium, Rainwater Charitable Foundation, Fort Worth, TX, USA  [3]UCLA, Department of Neurology, University of California Los Angeles, Los Angeles, CA, USA  [4]CHDI Management/CHDI Foundation, Princeton, NJ, USA

Correspondence: c.cariulo@irbm.com; ramee.lee@chdifoundation.org
*Lara Petricca and Andrea Caricasole contributed equally to this work
Lara Petricca's present address is Biopôle c/o Startlab, Epalinges, Switzerland
Andrea Caricasole's present address is Kedrion S.p.A., Bolognana, Italy

PTMs in HD models, identifying potential modulators of endogenous HTT phosphorylation (Cariulo et al, 2017, 2019). Among such modulators are kinases capable of increasing phosphorylation at HTT T3 or S13/S16 (Thompson et al, 2009; Bustamante et al, 2015; Bowie et al, 2018; Ochaba et al, 2019; Chiki et al, 2021; White et al, 2022) and pharmacological tools for proof-of-concept studies (Atwal et al, 2011; Alpaugh et al, 2017; Bowie et al, 2018; Cariulo et al, 2019). Efforts to identify candidate kinases and/or phosphatases associated with HTT phosphorylation at T3 and S13/S16 have so far led to the identification of IKBKB (inhibitor of nuclear factor kappa B kinase subunit beta) (Thompson et al, 2009; Bustamante et al, 2015; Ochaba et al, 2019), TBK1 (TANK-binding kinase 1) (Hegde et al, 2020), and PP1 (protein phosphatase 1) (Branco-Santos et al, 2017) as tool enzymes for T3 (IKBKB, PP1) and S13/S16 (IKBKB, TBK1) regulation. Of these, IKBKB and/or IKK (IκB kinase enzyme complex) signalling have been mechanistically associated with mutant HTT toxicity (Khoshnan et al, 2004, 2009, 2017; Thompson et al, 2009; Khoshnan & Patterson, 2011), *HTT* transcriptional regulation (Becanovic et al, 2015), and modulation of HTT S13 phosphorylation in vivo (Ochaba et al, 2019). This evidence, together with the known amenability to pharmacological modulation, makes IKK pathway kinases (and particularly IKBKB) strong tool candidates that can elucidate the mechanisms involved in regulating mutant HTT S13/S16 phosphorylation (Atwal et al, 2011; Ochaba et al, 2019).

Studies using phosphor-mimetic mutations or bona-fide HTT phosphorylation have however suggested that increased phosphorylation at T3 or S13/S16 would need to be achieved to decrease HTT toxicity (Atwal et al, 2007; Gu et al, 2009; Zheng et al, 2013; Cariulo et al, 2017; Chiki et al, 2017; DeGuire et al, 2018), thereby pointing at the need to develop ways to increase (rather than inhibit) the activity of relevant kinases. The paradoxical observations that IKBKB can directly phosphorylate HTT at S13/S16 (Thompson et al, 2009) and that IKBKB inhibitors can achieve the same effect (Atwal et al, 2011) suggest a level of complexity which requires further investigation, examining the role of the distinct IKK signalling pathways downstream of IKBKB in the regulation of HTT N17 phosphorylation. IKK signalling is among the better understood kinase signalling pathways and includes signalling by the IKK complex (IKBKB homodimers or IKBKB/IKBKA heterodimers in complex with IKBKG) regulating the activity of the transcription factor NF-κB (Hacker & Karin, 2006; Perkins, 2007; Israel, 2010; Yu et al, 2020), and non-IKK complex mediated signalling pathways, typically mediated by the non-canonical IKK kinases IKBKE (inhibitor of NF-κB kinase subunit epsilon) and TBK1, the best understood of which acts via the nuclear factors IRF3/IRF7 (interferon regulatory factor 3/7) (Fitzgerald et al, 2003; Sharma et al, 2003; Hacker & Karin, 2006; Clark et al, 2011; Balka et al, 2020; Yum et al, 2021). Although published evidence implicates IKBKB/IKBKA and canonical (NF-κB) mediated signalling in contributing to mutant HTT-mediated pathology and in *HTT* transcriptional regulation (Khoshnan & Patterson, 2011; Becanovic et al, 2015), little is known of the signalling downstream of IKBKB leading to HTT N17 phosphorylation.

Another level of complexity is provided by the presence of phosphorylation at T3 and S13/S16, which appear to functionally cross-talk at least in vitro (DeGuire et al, 2018). As IKBKB has been described capable of phosphorylating both these HTT phosphor-

epitopes (Thompson et al, 2009; Bustamante et al, 2015), it is important to investigate the effects of T3 and S13/S16 phosphorylation by IKBKB from the perspective of cross-talk, exploring how T3 modification affects S13/S16 phosphorylation and vice-versa.

Here, we have addressed these open questions using novel ultrasensitive assays for measuring endogenous levels of S13 HTT phosphorylation (pS13 HTT). We confirm, for the first time on endogenous HTT that IKBKB can quantitatively increase pS13 HTT levels in human cells and does so in a manner dependent on its catalytic activity, as determined by mutations inactivating its kinase domain and selective pharmacological inhibitors. We further uncover a role for phosphatases and in particular for PP2A in regulating the ability of IKBKB to modulate pS13 HTT levels. Interestingly, we found that the ability of IKBKB to increase pS13 HTT levels in human cells is shared with IKBKE but not with the other canonical IKK kinase, IKBKA, and used IKBKB mutants to define that monomeric and NEMO binding–incompetent IKBKB remain capable of increasing pS13 HTT levels, thus excluding a role for the IKK complex and canonical IKK signalling. Coherently, pS13 HTT modulation by IKBKB was found to be associated with activation of a non-canonical IKK pathway involving IRF3 activation rather than through IKBA (nuclear factor of kappa light polypeptide gene enhancer in B-cells inhibitor, alpha) activation (canonical IKK signalling). Functionally, the modulation of pS13 HTT levels by IKBKB is associated with a robust decrease of aggregation in a commonly used surrogate cell model of mutant HTT aggregation. Finally, we explored the role of residue cross-talk, providing the first evidence that T3 mutation affects basal and IKBKB induced pS13 HTT levels, an effect which translates functionally also in mutant HTT aggregation in cells. The identification of non-canonical IKK signalling as a regulator of HTT N17 phosphorylation and the finding that T3 and S13 phosphorylation may be subject to cross-talk represent major advances towards the understanding of approaches aiming at reducing mutant HTT pathology through increased N17 phosphorylation and provide novel insights towards the mechanistic understanding of N-terminal phosphorylation, regulation, and downstream pathophysiological implications.

## Results

### IKBKB modulation of endogenous pS13 HTT levels is regulated by okadaic acid–sensitive phosphatases

IKBKB can phosphorylate HTT N-terminal protein fragments in vitro (Thompson et al, 2009) and its overexpression can increase pS13 HTT levels in cells expressing HTT N-terminal fragments (Thompson et al, 2009; Bustamante et al, 2015). More recently, endogenous IKBKB was shown capable of regulating pS13 HTT levels in mice as determined by semiquantitative methods (Ochaba et al, 2019). Paradoxically, other studies showed that pharmacological inhibition of IKBKB leads to the same results in cells (Atwal et al, 2011). We sought to investigate this discrepancy by interrogating the role of phosphatases in regulating the effects of IKBKB on pS13 HTT levels. Phosphatases may affect pS13 HTT levels in different ways, including direct dephosphorylation of pS13 and regulation of the

catalytic activity, localization, and stability of S13 kinases and phosphatases. Of the different pharmacological tools available to probe phosphatase activity, okadaic acid (OA), a somewhat selective inhibitor of PP1 and PP2A (Dounay & Forsyth, 2002), has been previously used to demonstrate an increase in nuclear accumulation of N-terminal HTT fragments in cells associated with increased S16 HTT phosphorylation (Havel et al, 2011). Interestingly, both PP1 and PP2A have been associated with a role in regulating N-terminal HTT aggregation and phosphorylation (Metzler et al, 2010; Branco-Santos et al, 2017), whereas PP2A is a known regulator of IKBKB catalytic activity and downstream signalling (DiDonato et al, 1997). We therefore used OA to investigate the role of PP1/PP2A in regulating basal and IKBKB-induced pS13 HTT levels. For these mechanistic studies, we chose the human cell line HEK293T, being widely used as a reductionistic model to study HTT PTMs and their biology (Schilling et al, 2006; Cong et al, 2011; Zheng et al, 2013; Fodale et al, 2014; Bustamante et al, 2015; Huang et al, 2015; Daldin et al, 2017). This cell line expresses HTT protein endogenously at significant levels (Cariulo et al, 2017; Ratovitski et al, 2017), and the HTT protein is phosphorylated at T3 and S13/S16 under both endogenous and overexpression conditions (Cariulo et al, 2017, 2019), indicating the presence of the relevant signalling pathways required to phosphorylate HTT at these residues. The ability of overexpressed IKBKB or an IKBKB carrying a catalytically inactivating mutation K44M (Mercurio et al, 1997) to increase pS13 HTT levels in HEK293T cells overexpressing an N571 mutant HTT fragment was then assessed by Western blotting and the Singulex immunoassay (SMC assay) (Cariulo et al, 2019) in the presence or absence of OA at relevant concentrations (Havel et al, 2011) (Fig 1A and B, respectively). We assessed pS13 HTT levels by Western immunoblotting using the previously characterized rabbit polyclonal antibody, specific for pS13 HTT (Cariulo et al, 2019). As shown in Fig 1A, all HEK293T samples expressed N571 Q55 HTT protein at comparable levels. However, in the absence of little OA if any increase in pS13 HTT levels was observed in cells transfected with IKBKB as compared with cells transfected with the empty expression plasmid, in spite of robust expression of IKBKB. However, this IKBKB protein was slightly inactive as judged by the level of phosphorylation at residues S177/S181, which are auto-phosphorylation sites commonly used as a readout of IKBKB activity (Israel, 2010). We reasoned that the absence of robust pS13 HTT modulation by IKBKB was a result of low IKBKB activity, potentially due to the activity of endogenous phosphatases in HEK293T cells. In the presence of OA, however, a robust increase in pS13 HTT levels was observed in samples transfected with IKBKB but not in those transfected with an IKBKB kinase dead construct (Fig 1A), paralleled with a strong increase in pS177/pS181 IKBKB levels, indicative of robust kinase activity. Interestingly, basal levels of pS13 HTT, in the absence of transfected IKBKB, were not affected by OA treatment, suggesting that control of pS13 HTT levels by OA-sensitive endogenous phosphatases is not the predominant regulatory mechanism. When samples were analyzed by quantitative SMC assays, the capacity of OA to reveal a robust (ca. 10-fold), catalytic activity–dependent induction of pS13 HTT levels by IKBKB was confirmed (Fig 1B). We concluded that although basal levels of pS13 HTT in cells expressing N571 Q55 HTT fragments are not significantly altered by OA-mediated phosphatase inhibition, the capacity of

IKBKB to increase pS13 HTT levels is exquisitely controlled by OA-sensitive phosphatases, due to regulation of IKBKB activity, as evidenced by IKBKB pS177/pS181 levels. We next sought to determine if we could confirm the catalytic activity dependence of IKBKB in regulating pS13 HTT levels and the effect of OA using pharmacological rather than genetic means, on either overexpressed N571 Q55 HTT (Fig 1C and D) or on endogenously expressed HTT in HEK293T cells (Fig 1E and F). As shown in Fig 1C, N571 Q55 pS13 HTT levels were again not significantly modulated by IKBKB overexpression in the absence of OA, and addition of Bay-65-1942, a selective IKBKB inhibitor (Atwal et al, 2011), was similarly unable to significantly modulate these levels. In the presence of OA, the capacity of IKBKB to increase pS13 HTT levels was revealed and, importantly, it was sensitive to selective pharmacological inhibition by Bay-65-1942 in a concentration-dependent manner (Fig 1D). Substantially comparable results were obtained when the response of pS13 HTT levels expressed from the endogenous *HTT* locus to IKBKB and OA was investigated under the same conditions (Fig 1E and F), and the more quantitative SMC assay analysis on these samples confirmed the findings obtained by Western blotting (no OA: Fig 1G; OA treatment: Fig 1H). Therefore, we concluded that pS13 HTT levels, whether from the endogenous *HTT* locus or from an overexpressed HTT N-terminal fragment, could be robustly increased in response to IKBKB in a catalytic activity–dependent and phosphatase-sensitive manner. As OA is described as an inhibitor of PP1 and PP2A (Dounay & Forsyth, 2002) but exhibits greater selectivity towards the latter (Swingle et al, 2007), the role of PP2A on OA's ability to regulate IKBKB activity (S177/S181 auto-phosphorylation) and its effects on pS13 HTT levels was further evaluated using a specific siRNA to silence endogenous PP2A in HEK293T cells. As shown in Fig 2A, the PP2A silencing produced an increase in IKBKB auto-phosphorylation, indicative of increased IKBKB activity, concomitantly with increased pS13 HTT levels. The magnitude of the effects of PP2A silencing on pS13 HTT modulation by IKBKB was not comparable to that attained with OA, likely because of incomplete PP2A silencing obtained with RNAi and the difficulty of phenocopying pharmacological inhibition of a catalytic enzyme with RNAi (Weiss et al, 2007). Coherent with the observations made with OA, basal pS13 HTT levels (absence of IKBKB overexpression) were not affected by PP2A RNAi. Quantitative analysis of pS13 HTT levels of overexpressed N571 HTT bearing 55 CAG repeats (Q55) by SMC assay essentially confirmed the results obtained by Western blotting (Fig 2B). Consistent with a role for PP2A in regulating IKBKB activity, we were able to detect an interaction between overexpressed IKBKB and endogenously expressed PP2A in HEK293T cells by both immunoprecipitation (Fig 2C) and an ELISA interaction assay (Fig 2D). We next sought to determine if overexpression of PPP2CA, the catalytic subunit of PP2A, could affect the ability of IKBKB to influence pS13 HTT levels, an experiment which needed to be performed in the absence of OA and therefore against the background of endogenous inhibitory effects of PP2A on IKBKB. Using overexpression of N571 Q55 HTT and shorter transfection times (24 h) than those used in Fig 1, we were able to detect a reproducible, modest induction of pS13 HTT levels upon co-transfection with IKBKB, concomitant with significant S177/S181 IKBKB phosphorylation as a measure of IKBKB catalytic activity

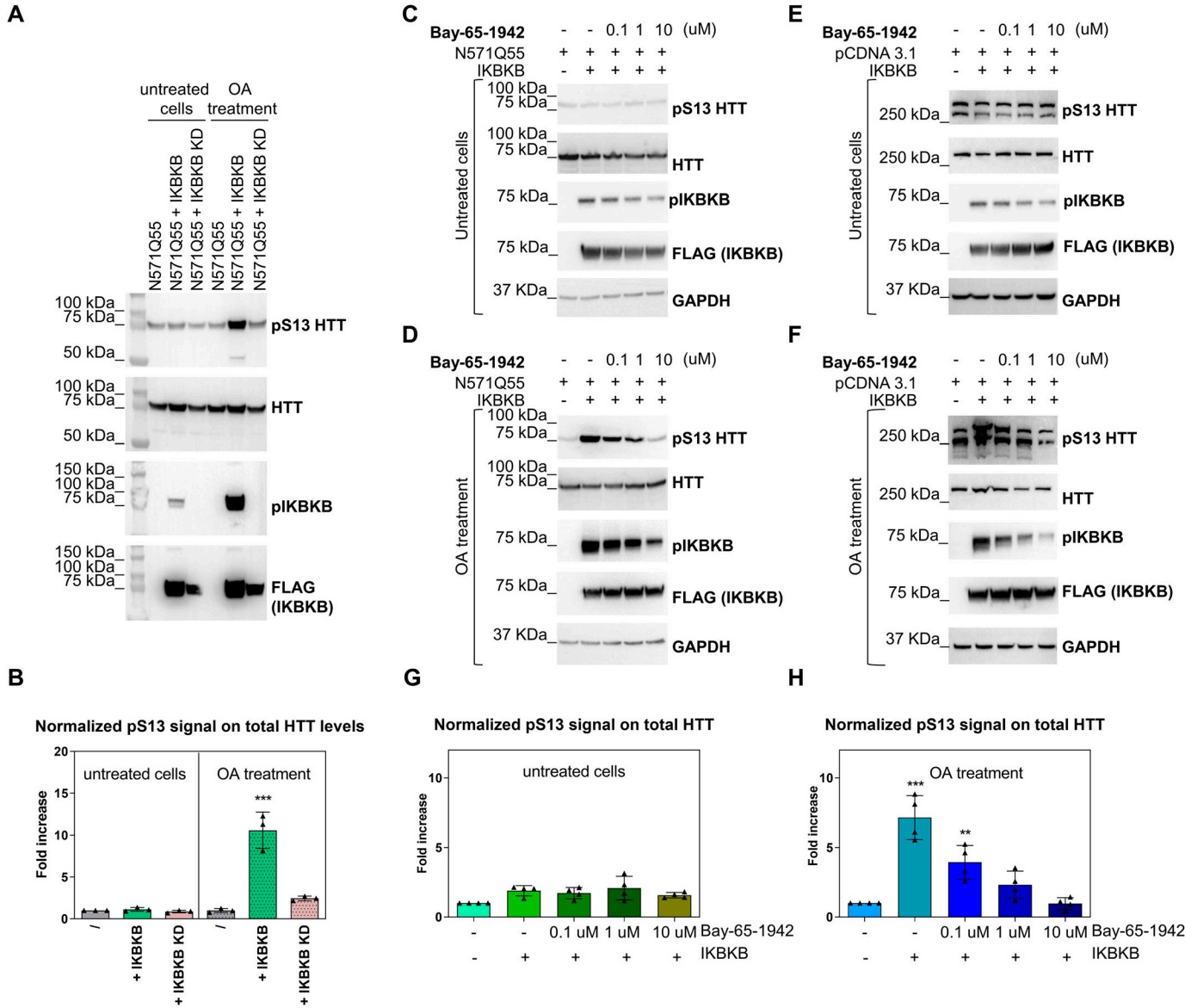

**Figure 1.   pS13 HTT levels are regulated by inhibitor of nuclear factor kappa B kinase subunit beta (IKBKB) and okadaic acid–sensitive phosphatases.**
**(A, B)** Treatment with okadaic acid (OA) of HEK293T cells overexpressing N571 Q55 HTT, with or without IKBKB or its kinase-dead mutant IKBKB KD. **(A)** Western blotting showing robustly increased pS13 HTT levels and increased auto-phosphorylation of IKBKB at residues S177/S181 upon IKBKB but not IKBKB KD overexpression in combination with OA treatment. Kinase expression and HTT levels were assessed using anti-FLAG and mAb 2166 antibodies, respectively. **(A, B)** Normalized pS13 signal on total HTT levels measured by Singulex immunoassay performed on the same lysates as in (A), confirming the Western blotting results. (Means and SDs were calculated on three biological replica. One-way analysis of variance, Dunnett's test [***$P$ < 0.0001]). **(C, D)** Western blotting of HEK293T cells overexpressing N571 Q55 HTT with or without IKBKB treated with IKBKB-inhibiting compound BAY-65-1942 at 0.1, 1, and 10 $\mu M$, without OA (C) and in combination with OA (D), demonstrating the capacity of IKBKB to increase pS13 HTT levels only in the presence of OA, and of BAY-65-1942 compound to inhibit IKBKB's activity on pS13 in a dose-dependent way. IKBKB expression and its auto-phosphorylation were assessed using anti-FLAG and anti-pIKBKB antibodies, respectively. Protein loading and HTT levels were assessed using anti-GAPDH and mAb 2166 antibodies, respectively. **(C, D, E, F)** Same experimental conditions as in (C, D) evaluated on endogenous HTT in HEK293T cells confirming the overexpression results without OA (E) or in presence of OA (F). IKBKB expression and its auto-phosphorylation were assessed using anti-FLAG and anti-pIKBKB antibodies, respectively. Protein loading and HTT levels were assessed using anti-GAPDH and mAb 2166 antibodies, respectively. **(C, D, E, F, G, H)** Singulex immunoassay on same lysates as in (C, D, E, F) (means and SDs of n = 2 overexpressed HTT merged with n = 2 endogenous HTT experiments) without OA (G) and in combination with OA (H), confirming Western blotting results. (Means and SDs were calculated on four biological replica. One-way analysis of variance, Dunnett's test [**$P$ < 0.001; ***$P$ < 0.0001]).

(Fig 2E and F). Under these conditions, co-transfection of PPP2CA effectively abolished IKBKB-induced phosphorylation at S177/S181 and pS13 HTT levels, consistent with the observations made using RNA knock-down. Collectively, the data suggest that the capacity of IKBKB to influence pS13 HTT levels in HEK293T cells is dependent on its catalytic activity that the catalytic activity of IKBKB is regulated by OA-sensitive phosphatases and that PP2A contributes to IKBKB inducible, but not basal, pS13 HTT levels. These findings are fully consistent with the known role of PP2A in regulating IKBKB activity (Barisic et al, 2008; Tsuchiya et al, 2017).

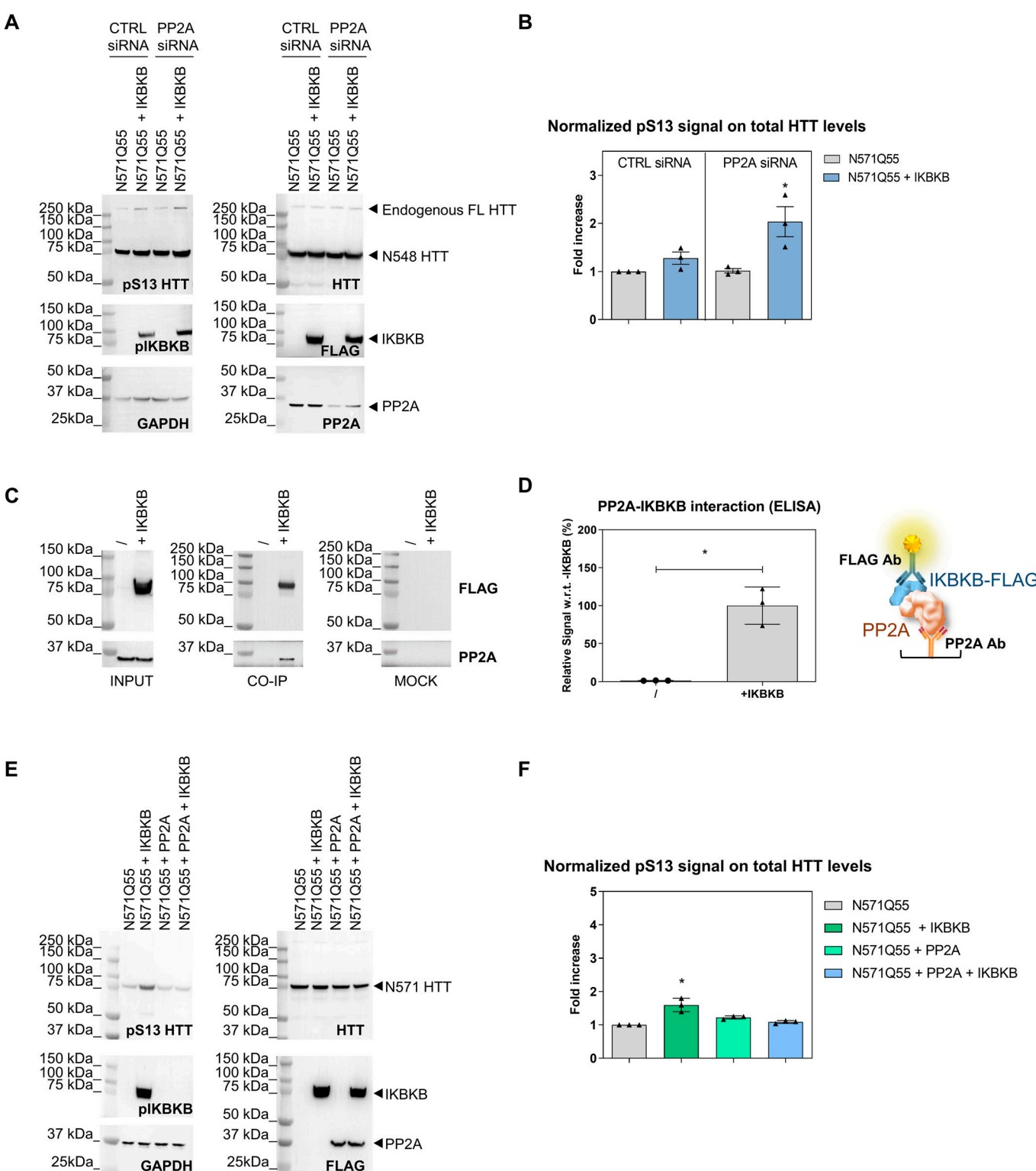

**Figure 2. Protein phosphatase 2A (PP2A) interferes on inhibitor of nuclear factor kappa B kinase subunit beta's (IKBKB's) capacity to phosphorylate HTT S13.**
**(A, B)** Endogenous PP2A knockdown consequent to specific siRNA treatment in HEK293T cells transiently co-transfected with N571 Q55 HTT and IKBKB produced an increased response to IKBKB compared with siRNA control. **(A)** Western blotting showing that IKBKB overexpression leads to higher phosphorylation on HTT S13 when PP2A is silenced (as shown by anti-PP2A Ab). Increased auto-phosphorylation of IKBKB on S177/S181 as a result of endogenous PP2A knockdown. Protein loading, kinase and phosphatase expression, and HTT levels were assessed using anti-GAPDH, anti-FLAG, anti-PP2A, and mAb 2166 antibodies, respectively. **(A, B)** Normalized pS13 signal on total HTT levels measured by Singulex immunoassay performed on the same lysates as in (A) confirming Western blotting results. (Means and SDs were calculated on three biological replica. One-way analysis of variance, Dunnett's test [*$P < 0.05$]). **(C, D)** Endogenous PP2A in HEK293T cells interacts with IKBKB. **(C)** Western

## IKBKB modulation of endogenous pS13 HTT levels involves a non-canonical IKK pathway associated with IRF3 signalling

IKK kinases have been extensively studied owing to their involvement in multiple areas of human disease, including inflammation, immunity, and cancer (Mercurio et al, 1997; Zandi et al, 1997; Sharma et al, 2003; Hacker & Karin, 2006; Perkins, 2007; Israel, 2010; Clark et al, 2011). Several functional mutants have been extensively characterized to probe the involvement of IKBKB in IKK pathways, which broadly can be grouped into IKK complex–mediated (canonical) and non-canonical. Canonical IKK signalling involves the formation of a functional complex involving IKBKB, IKBKA, and IKBKG/NEMO and results in IKBA phosphorylation and degradation, release of NF-κB from its inhibitor IKBA, and translocation of NF-κB to the nucleus. NF-κB executes its role as a transcription factor (Israel, 2010; Yu et al, 2020), regulating the expression of a number of genes involved in, among others, inflammation and also HTT itself (Becanovic et al, 2015). In contrast, non-canonical IKK signalling is largely IKBKG independent and includes pathways of which the best understood are those triggered by the non-canonical IKK kinases, TBK1, and IKBKE, which involve IRF (IRF3/7) rather than NF-κB as transcriptional effectors (Fitzgerald et al, 2003; Hacker & Karin, 2006; Balka et al, 2020; Yum et al, 2021). A range of IKBKB mutants have been generated and characterized which can be leveraged to genetically investigate the mechanism of IKBKB regulation of pS13 HTT levels. The HEK293T cell model, with its simplicity and amenability to manipulation, and the observation that OA unmasks the ability of IKBKB to modulate pS13 HTT by inhibiting endogenous phosphatases including PP2A, provides an appropriate environment for these mechanistic investigations. Further validation and hypothesis building, with regard to relevance for HD needs to be subsequently tested in other, more complex contexts. Equipped with IKBKB and a set of IKBKB mutants (Fig 3A) including a catalytically inactive (IKBKB KD) variant (Mercurio et al, 1997), an IKBKG/NEMO binding–incompetent (IKBKB NBD) mutant (May et al, 2002), a variant with decreased kinase activity through phosphor-mimetic mutations at residues S177/S181 (IKBKB S177E/S181E) (Mercurio et al, 1997; Huynh et al, 2000; Kishore et al, 2002; Liu et al, 2013; Polley et al, 2013) and an IKBKB mutant incapable of dimerization (IKBKB LZ) (Hauenstein et al, 2014), we set out to dissect the mechanisms through which IKBKB increases pS13 HTT levels in the HEK293T model. All mutants were C-terminally FLAG-tagged to facilitate comparative analysis of expression levels. We co-transfected IKBKB and the indicated mutants in HEK293T cells together with an expression plasmid encoding N571 Q55 HTT, in the presence or absence of OA, and determined their activity on pS13 HTT levels. As

shown in Fig 3B, in the absence of OA treatment, IKBKB did not produce a robust increase in pS13 HTT levels, as previously observed, as did none of the IKBKB mutants in spite of varying expression levels of these latter (see anti-FLAG blotting). Interestingly, in the presence of OA, IKBKB, and IKBKB mutant incapable of forming a complex with IKBKG/NEMO (IKBKB NBD) or the IKBKB mutant incapable of dimerization (IKBKB LZ) were still capable of increasing pS13 HTT levels, with a concomitant increase in S177/S181 phosphorylation, consistent with an IKK complex (IKBKG/IKBKA)–independent mechanism. This was evident despite the disparity in expression levels of the different constructs, which became evident with the kinase-dead IKBKB (IKBKB KD) and the IKBKG/NEMO binding–incompetent IKBKB (IKBKB NBD). The latter construct still produced a substantial increase in pS13 HTT levels, in spite of significantly lower expression levels with respect to for instance IKBKB. These samples were also analyzed using SMC assays, producing essentially comparable results to those obtained by Western blotting (Fig 3C). The data indicated that, in HEK293T cells, IKBKB regulates pS13 HTT levels through a non-canonical, IKK complex–independent mechanism, dependent on the catalytic activity of IKBKB which itself is regulated by protein phosphatases including PP2A. We sought to confirm and further extend these findings by analyzing the effects of overexpression of IKBKB and its mutants on endogenous pS13 HTT levels in comparison with overexpression of a canonical (IKBKA) or a non-canonical (IKBKE) IKK kinase and of the non-catalytic scaffolding component of the canonical IKK complex, IKBKG/NEMO. As shown in Fig 4A, in the absence of OA-overexpressed IKBKB, IKBKB NBD, and IKBKB LZ were able to modestly increase endogenous pS13 HTT levels, whereas IKBKB KD and the IKBKB S/E mutant (decreased kinase activity) were not, as observed in cells expressing the N571 Q55 HTT protein (Fig 3B and C). Interestingly, overexpression of the non-canonical kinase IKBKE but not of the canonical kinase IKBKA or of IKBKG, the non-catalytic scaffolding component of the canonical IKK complex, was also able to increase endogenous pS13 HTT levels (Fig 4A and B). In the presence of OA, these effects were greatly enhanced, as expected (Fig 4C and D). These results confirmed that non-canonical IKK signalling, but not signalling mediated by the canonical IKK complex, was responsible for regulating pS13 HTT levels. To further support this finding, we examined the phosphorylation of IRF3, a characterized effector of non-canonical IKK signalling, which upon phosphorylation by non-canonical IKK kinases is activated and acts as the transcriptional mediator of the pathway (Fitzgerald et al, 2003; Hacker & Karin, 2006). Indeed, we observed complete correlation between pS13 HTT phosphorylation and IRF3 phosphorylation (Fig 4A and C). Collectively, the effects of IKBKB on pS13

blotting of HEK293T cells lysates overexpressing IKBKB FLAG–tagged (INPUT) pulled-down by anti-FLAG antibody (CO-IP) or unrelated anti-glial fibrillary acidic protein antibody (MOCK), and revealed by anti-FLAG and anti-PP2A antibodies. Anti-FLAG and anti-PP2A signals in CO-IP at the expected molecular weights demonstrate the PP2A–IKBKB interaction. **(D)** ELISA assay using anti-PP2A as capture antibody and anti-FLAG as detection antibody performed on HEK293T cell lysates transfected with/without IKBKB FLAG–tagged confirms the interaction between endogenous PP2A and IKBKB. (Means and SDs were calculated on three biological replica. T test [*P < 0.05]). **(E, F)** IKBKB's capacity to influence pS13 HTT levels is affected by PP2A. **(E)** Western blotting of HEK293T cells overexpressing N571 Q55 HTT with or without IKBKB FLAG–tagged, PP2A FLAG–tagged or the combination of both. The lack of up-regulation of pS13 HTT levels by IKBKB when PP2A is overexpressed demonstrates that PP2A can regulate IKBKB's activity (as confirmed by the absence of IKBKB auto-phosphorylation in the same condition) rather than dephosphorylating HTT S13 (as shown by the lack of modulation of pS13 HTT levels in the presence of PP2A). Protein loading, kinase and phosphatase expression, and HTT levels were assessed using anti-GAPDH, anti-FLAG, and mAb 2166 antibodies, respectively. **(E, F)** Normalized pS13 signal on total HTT levels measured by Singulex immunoassay, performed on the same lysates as in (E) confirming the Western blotting results. (Means and SDs were calculated on three biological replica. One-way analysis of variance, Dunnett's test [*P < 0.05]).

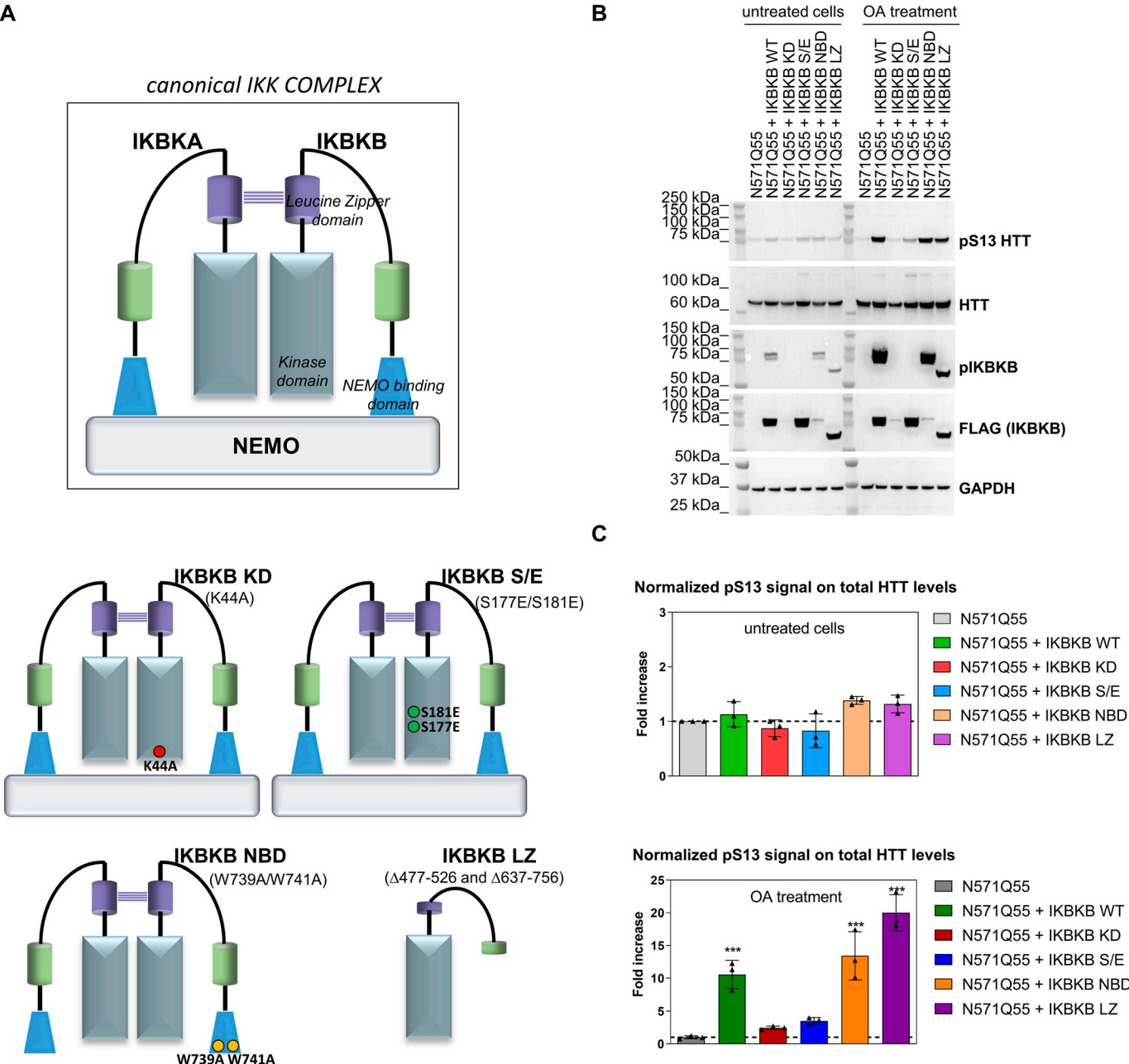

**Figure 3. Inhibitor of nuclear factor kappa B kinase subunit beta (IKBKB) effects on pS13 HTT levels are independent of IKBKB's activity in the canonical IKK complex.**
**(A)** Schematic representation of IKBKB mutant constructs used here. **(B, C)** Western blotting of HEK293T cells overexpressing N571 Q55 HTT with or without IKBKB WT and IKBKB mutants, in presence or absence of OA treatment. pS13 HTT levels increased only in the presence of the IKBKB WT, the IKBKB NBD, and the IKBKB LZ mutants when cells were treated with OA, indicating that the canonical IKK pathway is not involved in regulation of pS13 by IKBKB. Protein loading, kinases expression and activity, and HTT levels were assessed using anti-GAPDH, anti-FLAG, anti-pIKBKB, and mAb 2166 antibodies, respectively. **(B, C)** Normalized pS13 signal on total HTT levels measured by Singulex immunoassay, performed on the same lysates as in (B) confirming the Western blotting results. (Means and SDs were calculated on three biological replica. One-way analysis of variance, Dunnett's test [***$P < 0.0001$].)

HTT levels expressed from the endogenous *HTT* locus or from overexpressed HTT fragments are dependent on its catalytic activity, which is regulated by phosphatases including PP2A, can be phenocopied by non-canonical IKK kinases such as IKBKE but not by canonical IKK kinase IKBKA and, consistently, are mediated by a non-canonical IKK pathway associated with IRF3 activation.

IKBKB is known to directly phosphorylate purified, isolated HTT exon 1 protein in vitro (Thompson et al, 2009) but no evidence of a direct interaction between IKBKB and HTT has been demonstrated in cellular or animal models. We therefore asked whether the kinase exerted its activity on S13 in a direct way, through interaction with HTT in a cellular context, by applying a

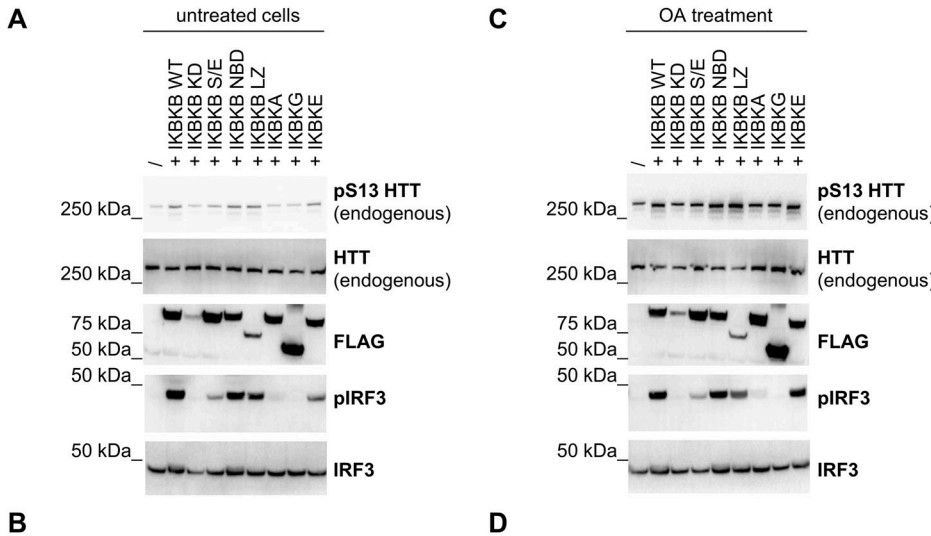

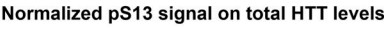

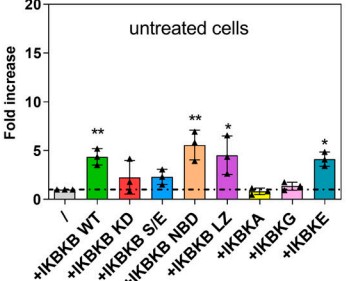

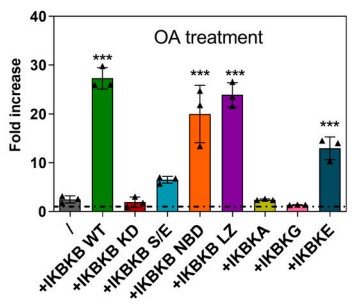

**Figure 4. Inhibitor of nuclear factor kappa B kinase subunit beta (IKBKB) regulation of endogenous pS13 HTT levels is associated with an interferon regulatory factor 3 (IRF3) mediated, non-canonical IKK pathway.**
**(A, B)** HEK293T cells overexpressing IKBKB WT and IKBKB mutants, IKBKA, IKBKG, and inhibitor of nuclear factor kappa-B kinase subunit epsilon (IKBKE) in absence of OA. **(A)** Western blotting showing a modest increase of endogenous pS13 HTT levels upon expression of IKBKB WT, IKBKB NBD mutant, IKBKB LZ mutant and of the non-canonical IKBKE kinase. The non-canonical IRF3 pathway is involved in pS13 HTT regulation as demonstrated by the increase of endogenous phosphor-IRF3. Kinases expression, HTT levels, and endogenous IRF3 expression were assessed using anti-FLAG, mAb 2166, and anti-IRF3 antibodies, respectively. **(A, B)** Normalized pS13 signal on total HTT levels measured by Singulex immunoassay, performed on the same lysates as in (A) confirming the Western blotting results. (Means and SDs were calculated on three biological replica. One-way analysis of variance, Dunnett's test [*$P < 0.05$; **$P < 0.001$]). **(C, D)** HEK293T cells overexpressing IKBKB WT and IKBKB mutants, IKBKA, IKBKG, and IKBKE in the presence of OA. **(A, B, C)** Western blotting showing that the presence of OA greatly enhances the effects described in (A) and (B), coherent with effects on pS13 HTT being dependent on the catalytic activity of IKBKB. **(D)** Normalized pS13 signal on total HTT levels measured by Singulex immunoassay, performed on the same lysates as in (C) confirming the Western blotting results. (Means and SDs were calculated on three biological replica. One-way analysis of variance, Dunnett's test [***$P < 0.0001$]).

quantitative protein–protein interaction assays (TR-FRET and ELISA) to HEK293T cells co-expressing N571 Q55 HTT construct with or without S13A/S16A phosphor-abrogative mutations and IKBKB. As shown in Fig 5A–C, IKBKB was able to interact with HTT N571 Q55 protein and this interaction was independent of the phosphorylation status of S13, as the abrogation of S13 phosphorylation (S13A/S16A mutant) did not impact a direct interaction. On the other hand, the interaction was dependent on the catalytic activity of IKBKB, as the catalytically inactive mutant (KD) interacted with the HTT protein to a lesser extent (Fig 5D). Based on these data, we conclude that IKBKB can interact with HTT in this cell context, and that this interaction is dependent on the integrity of the catalytic domain of IKBKB but independent of IKBKB's HTT substrate phosphorylation.

### IKBKB influences mutant HTT aggregation through increased S13 HTT phosphorylation and cross-talk between T3 and S13

One of the key functional consequences of polyQ expansion believed to contribute to HD pathology is misfolding and aggregation of short (exon 1 encoded) N-terminal HTT fragments, produced either from misplicing and/or proteolysis (Bates, 2003; Landles et al, 2010; Weiss et al, 2012; Gipson et al, 2013; Sathasivam et al, 2013; Bates et al, 2015; Saudou & Humbert, 2016; Neueder et al, 2017; Franich et al, 2019). Although mutant HTT aggregation is invariably

(but to different extents) detectable in HD mouse models (William Yang & Gray, 2011; Menalled & Brunner, 2014), this process is not easily reproduced in cellular models of full-length mutant HTT. To address this issue, expression of the exon 1 fragment of mutant HTT in cells, often tagged with fluorescent proteins, is generally used as a surrogate mutant HTT aggregation model (Cooper et al, 1998; Li & Li, 1998; Atwal et al, 2007; Olshina et al, 2010; Ross & Tabrizi, 2011; Zheng et al, 2013; Bates et al, 2015; Li et al, 2016; Branco-Santos et al, 2017). Indeed, mutant HTT aggregation and toxicity in cellular and animal models of HD is robustly affected by phosphor-mimetic mutations at S13/S16 and by other mutations within the N17 domain (Atwal et al, 2007; Gu et al, 2009; Greiner & Yang, 2011; Zheng et al, 2013; Branco-Santos et al, 2017). Although some pharmacological tools demonstrated to increase pS13/pS16 HTT levels have shown benefit in animal and cellular models of HD, modifying different phenotypes including mutant HTT aggregation (Di Pardo et al, 2012; Alpaugh et al, 2017; Bowie et al, 2018), the lack of suitable direct genetic modulators of pS13/pS16 HTT levels have so far limited the capacity to determine if effects of phosphor-mimetic mutations are indeed a true phenocopy of increased phosphorylation. Together with the data presented here, the identification of IKBKB as a kinase capable of increasing S13/S16 HTT phosphorylation (Thompson et al, 2009) and the evidence supporting its role as a physiologically relevant modulator of N-terminal HTT phosphorylation (Ochaba et al, 2019) points to IKBKB as a suitable tool to probe

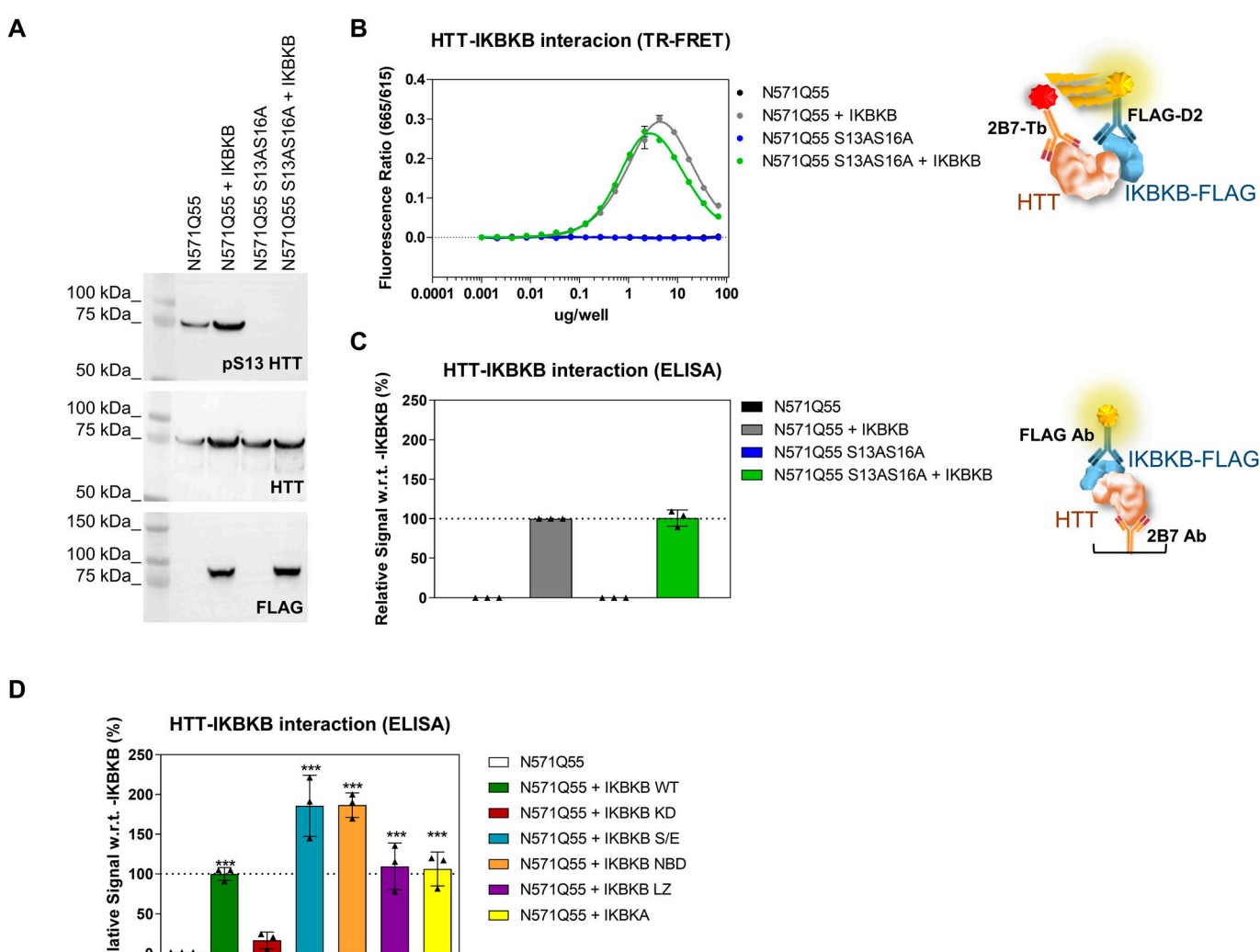

**Figure 5.    Inhibitor of nuclear factor kappa B kinase subunit beta (IKBKB) interacts with HTT in cells in a manner dependent on its catalytic activity but independent of HTT phosphorylation at S13.**

**(A, B, C)** Western blotting of HEK293T cells overexpressing N571 Q55 HTT WT or S13A/S16A phosphor-abrogative mutant with or without IKBKB FLAG–tagged. pS13 HTT levels, IKBKB expression, and HTT levels were assessed using anti-pS13, anti-FLAG, and mAb 2166 antibodies, respectively. **(B)** TR-FRET protein–protein interaction assay using 2B7-Tb/anti-FLAG–D2 antibody pair showing a strong interaction of N571 Q55 HTT WT not only with IKBKB but also with the phosphor-abrogative N571 Q55 HTT S13A/S16A mutant. **(B, C)** ELISA protein–protein interaction assay using 2B7 as capture and anti-FLAG as detection antibodies used as orthogonal readout confirming the data in (B). **(D)** ELISA protein–protein interaction assay of HEK293T cells overexpressing N571 Q55 HTT WT and IKBKB WT or IKBKB KD, IKBKB S/E, IKBKB NBD, IKBKB LZ, or IKBKA, showing that the interaction of N571 Q55 HTT with IKBKB is robustly affected only in the presence of the kinase-dead mutation and therefore this interaction is dependent on the integrity of IKBKB's catalytic domain. (Means and SDs were calculated on three biological replica. One-way analysis of variance, Dunnett's test [***$P < 0.0001$]).

the role of increased phosphorylation on mutant HTT aggregation. To do this, we first implemented a surrogate cellular model of mutant HTT aggregation, based on overexpression of HTT exon 1 bearing an expanded polyQ (Q72) repeat, C-terminally fused to GFP, in HEK293T cells (Fodale et al, 2014). To validate the model, we sought to confirm the previously reported effects of N17 mutations on mutant exon 1 HTT (mut Ex1 HTT) aggregation, where phosphor-mimetic mutations at S13/S16 resulted in robustly decreased aggregation (Atwal et al, 2007; Greiner & Yang, 2011; Branco-Santos et al, 2017), using previously validated and characterized constructs (Cariulo et al, 2017; Fodale et al, 2017). As illustrated in Fig 6A and B, consistent with published data (Gu et al, 2009; Branco-Santos et al, 2017) S13A/S16A and S13D/S16D double mutations produced

dramatic and opposite effects on mut Ex1 HTT aggregation (negative controls are shown in Fig S1). These data are in agreement with the proposed role for phosphomimesis in decreasing mut Ex1 HTT aggregation (Greiner & Yang, 2011). Conversely, T3A and T3D mutations also decreased aggregation (Fig 6), but to the same extent indicative that the effect cannot be reconciled with phospho-mimesis, contrary to what observed for S13/S16. In agreement with the results obtained using the independent T3D and S13D/S16D mutations, the triple-T3D/S13D/S16D mutant produced a robust decrease in aggregates. Interestingly, a triple-T3A/S13A/S16A mutation resulted in increased aggregation, comparable to that attained by the double S13A/S16A mutation (Fig 6), suggesting that the beneficial effect of the T3A mutation on aggregation

## A

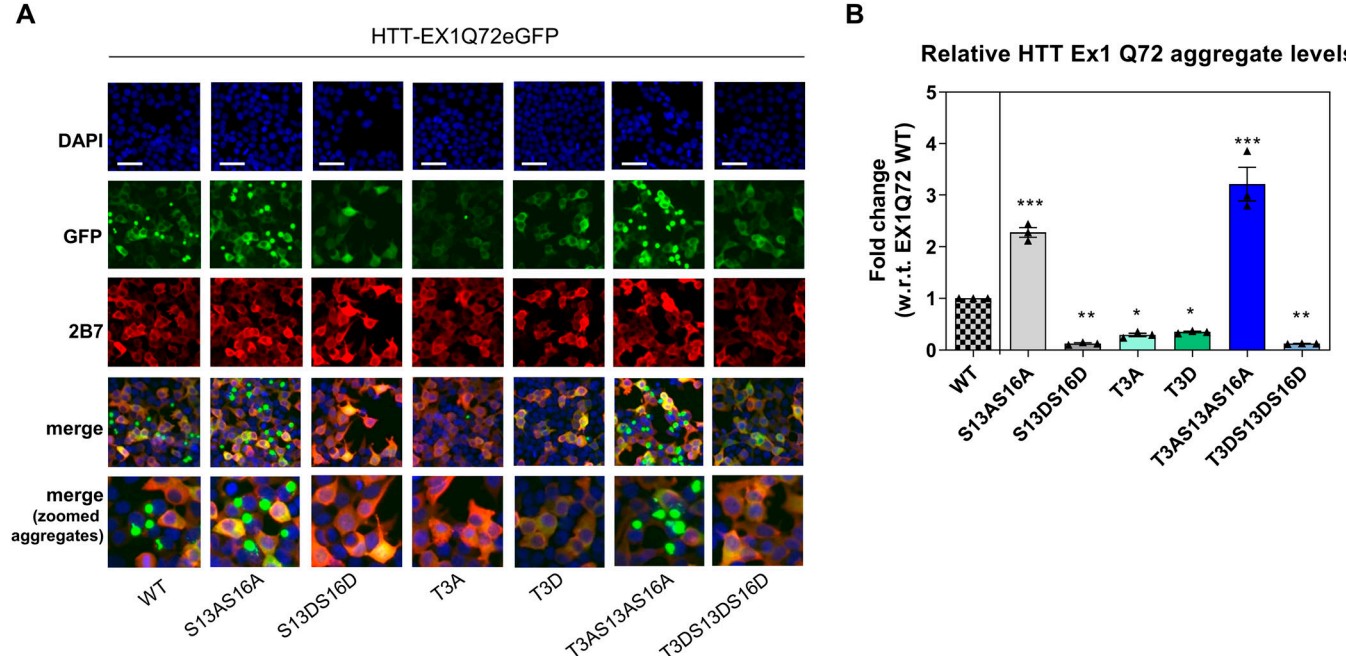

**Figure 6. HTT N-terminal phosphor-abrogative/mimetic mutations affect mutant HTT aggregation.**
**(A)** Immunofluorescence of HEK293T cells overexpressing HTT Ex1 Q72-EGFP WT or phosphor-mutants (S13A/S16A, S13D/S16D, T3A, T3D, T3A/S13A/S16A, T3D/S13D/S16D), showing increased HTT aggregates in the presence of the phosphor-abrogative S13A/S16A mutation and decreased aggregation in the presence of the phosphor-mimetic S13D/S16D mutation. T3A and T3D mutations produced a comparable decrease in HTT aggregation. Expression of a triple phosphor-abrogative mutant T3A/S13A/S16A resulted in increased aggregation to a comparable degree to the double-S13A/S16A mutation, whereas the respective phosphor-mimetic triple mutant (T3D/S13D/S16D) led to the opposite result. Nuclear signal, aggregates signal, and HTT levels were assessed by DAPI staining, GFP-signal acquisition, and 2B7 antibody staining, respectively. (Representative images of n = 3 independent experiments. Scale bars represent 50 μm). B Quantitative analysis of images in A. (Means and SDs were calculated on three biological replica. One-way analysis of variance, Dunnett's test [*$P < 0.05$; **$P < 0.001$; ***$P < 0.0001$]).

requires the presence of intact S13/S16 residues. One possibility is that mutation of T3 results in increased phosphorylation of S13/S16 and that the decreased aggregation observed with T3 mutation at least in part is mediated by increased pS13/pS16 levels. To investigate this aspect, we measured basal and IKBKB-induced pS13 HTT levels by densitometry analysis of Western blotting in cells transfected with HTT Ex1 Q72-EGFP and with T3A or S13A/S16A mutations (Fig 7A and B). As previously observed, a modest effect of IKBKB co-transfection on pS13 HTT levels was observed in cells transfected with the HTT Ex1 Q72-EGFP construct, owing to the absence of OA treatment. Indeed, the presence of a T3A mutation significantly increased basal pS13 HTT levels, by about fivefold and correspondingly potentiated the effects of IKBKB on pS13 HTT levels relatively to the protein expressed from the unmodified construct (Fig 7A and B). We also measured pT3 HTT levels to determine if S13/S16 modification would affect pT3 HTT levels, which was not the case (Fig 7A and C), however confirming the ability of IKBKB to increase pT3 HTT levels (Bustamante et al, 2015). Therefore, in this reductionistic, surrogate model of mutant HTT aggregation, we were able to confirm previous reports of the capacity of phosphor-mimetic S13/S16 mutations to reduce aggregation in cells, pointing to the role for T3 in regulating basal- and IKBKB-induced pS13 HTT levels. This also provides an explanation for the observed effects of T3A mutations to reduce aggregation in an

S13/S16-dependent manner (Fig 6). We next proceeded with examining the role of IKBKB-regulated pS13 HTT levels in the aggregation process. As previously shown, IKBKA and IKBKB S/E co-expression did not modulate pS13 HTT levels and as reported in Fig 8A and B, also failed to modulate aggregation whereas IKBKB robustly reduced HTT Ex1 Q72-EGFP aggregation in a catalytic-dependent manner, demonstrated by lack of effects of IKBKB KD. As this effect could be mediated directly through N17-phosphorylation-dependent or -independent mechanisms, we interrogated the relative importance of S13/S16 and T3 for the IKBKB-mediated reduction in mutant HTT aggregation using the available mutations. As illustrated in Fig 8C and D, the effects of IKBKB on HTT Ex1 Q72-EGFP aggregation were exquisitely dependent on the presence of intact S13/S16 residues, with T3 mutation further enhancing the capacity of IKBKB to influence the aggregation process, although T3A HTT Ex1 Q72-EGFP showed an already reduced propensity to aggregate (Fig 6). We conclude that, in this surrogate cellular model of mutant HTT aggregation, IKBKB can reduce a pathologically relevant phenotype in a manner dependent on its catalytic activity and on the presence of intact S13/S16 residues, which directly links the ability of IKBKB to regulate pS13 HTT levels with its capacity to reduce aggregation. To our knowledge, this is the first demonstration that increasing S13 phosphorylation phenocopies the effects of S13/S16 phosphor-mimetic mutations.

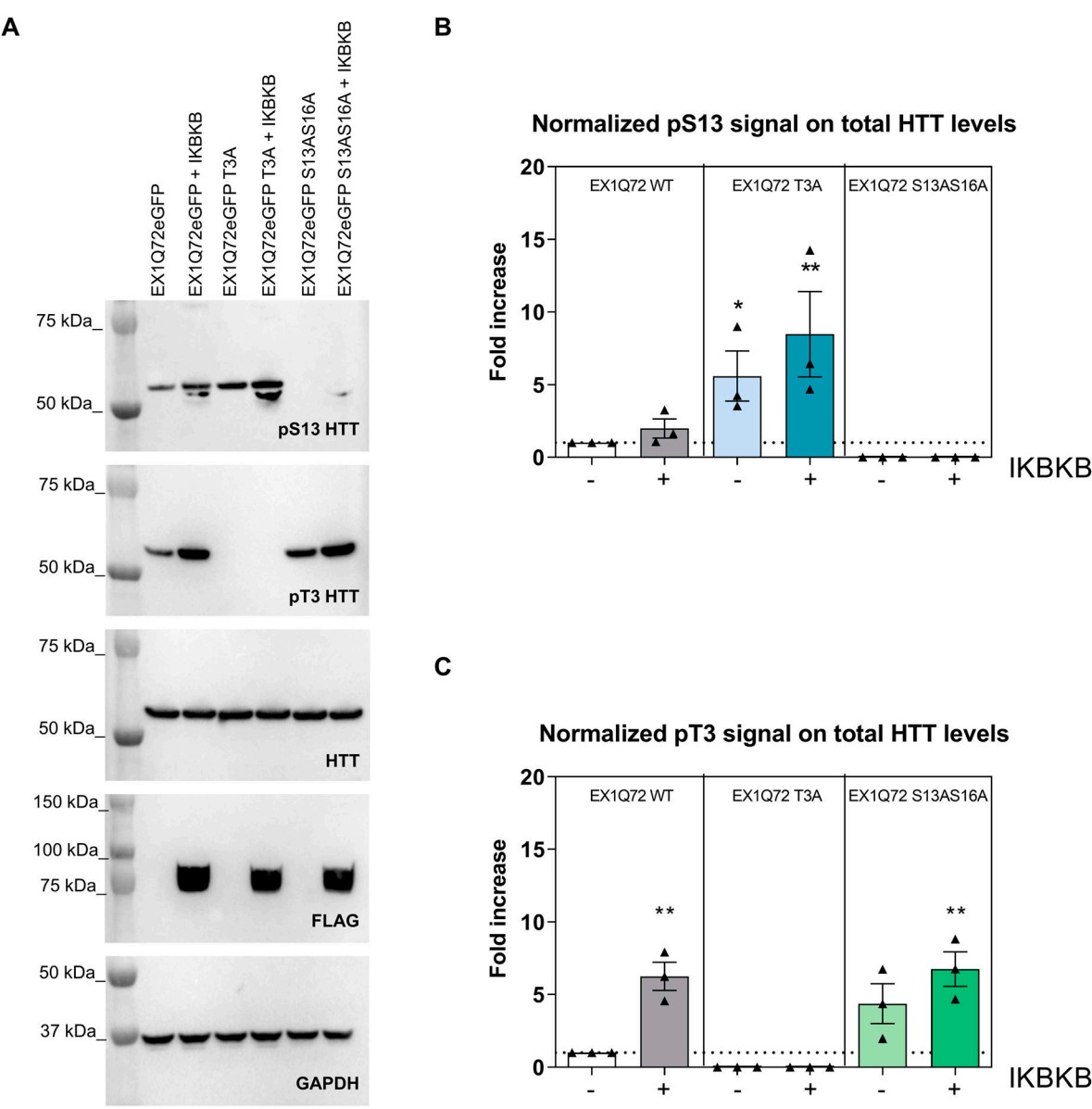

**Figure 7. Inhibitor of nuclear factor kappa B kinase subunit beta (IKBKB) influences pS13 HTT levels through cross-talk between T3 and S13.**
**(A)** Western blotting of HEK293T cells overexpressing HTT Ex1 Q72-EGFP WT or phosphor-abrogative mutants (T3A or S13A/S16A) with or without IKBKB. Protein loading, pS13 HTT levels, pT3 HTT levels, IKBKB expression, and HTT levels were assessed using anti-GAPDH, anti-pS13, anti-pT3, anti-FLAG, and 4C9 antibodies, respectively. **(A, B, C)** Densitometry analysis of Western blotting in (A) showing normalized pS13 HTT levels on total HTT (B) and normalized pT3 HTT levels on total HTT (C), demonstrating that the presence of T3A mutation increases basal pS13 HTT levels and enhances the effects of IKBKB on pS13 HTT levels. (Means and SDs were calculated on three biological replica. One-way analysis of variance, Dunnett's test [*$P < 0.05$; **$P < 0.001$]).

## Discussion

The role of HTT PTMs in ameliorating pathological phenotypes in HD mouse models is well supported by the findings that phosphor-mimetic mutations at residues associated with phosphorylation (e.g., S13/S16 and S421) can significantly affect mutant HTT biology (Gu et al, 2009; Kratter et al, 2016; Xu et al, 2020). Recent discoveries of pharmacological modulators of kinase activity (Simpson et al, 2009; Ferguson & Gray, 2018), open the possibility of developing candidate tools to better increase our understanding of the

mechanisms involved in modulating HTT phosphorylation and their contribution to HD pathophysiology.

However, interesting conceptually, detailed studies assessing their potential therapeutic relevance has been hampered by the lack of specific and sensitive technologies to adequately determine the biophysical properties and biologically dynamic nature of bona fide PTMs. Where pharmacological modulators have been identified (Marion et al, 2014; Alpaugh et al, 2017; Bowie et al, 2018), their efficacy, potency, selectivity, precise mechanism of action, and amenability as tools for further proof-of-concept

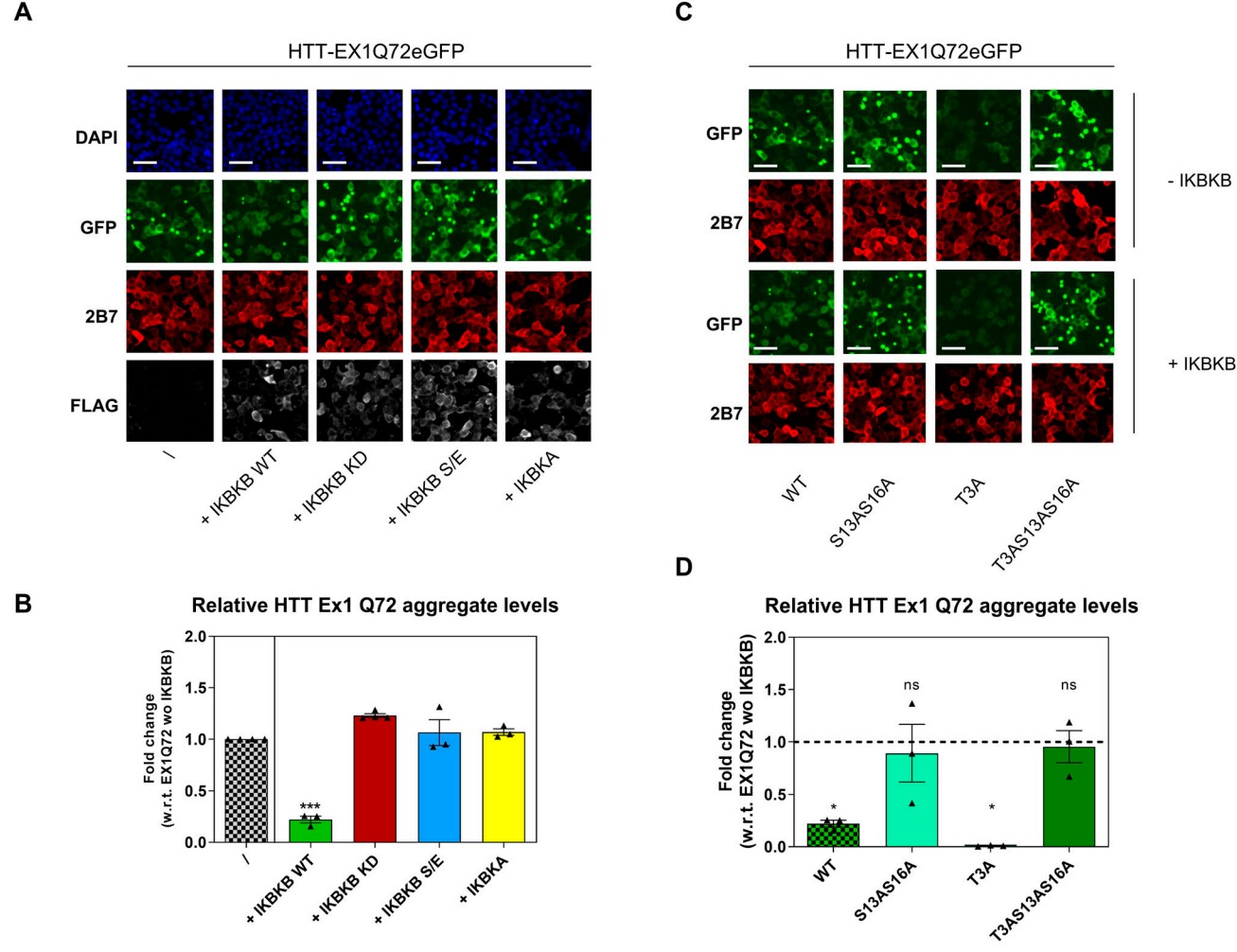

**Figure 8. Inhibitor of nuclear factor kappa B kinase subunit beta (IKBKB) influences mutant HTT aggregation dependent on an intact catalytic site and on the presence of S13/S16 residues.**

**(A, B)** HEK293T cells overexpressing HTT Ex1 Q72-EGFP WT and IKBKB WT or IKBKB KD, IKBKB S/E or IKBKA. **(A)** Immunofluorescence showing decreased mutant HTT aggregation only in the presence of WT IKBKB kinase. Aggregation levels are not modulated in the presence of the KD mutation, S/E mutation, or in the presence of IKBKA. Nuclear signal, aggregates signal, HTT levels, and kinases expression were assessed by DAPI staining, GFP-signal acquisition, 2B7 antibody, and anti-FLAG antibody staining, respectively. (Representative images of n = 3 independent experiments. Scale bars represent 50 $\mu$m). **(A, B)** Quantitative analysis of images in (A). (Means and SDs were calculated on at least three biological replica. One-way analysis of variance, Dunnett's test [***$P$ < 0.0001]). **(C, D)** HEK293T cells overexpressing HTT Ex1 Q72-EGFP WT or phosphor-abrogative mutants (S13A/S16A, T3A, T3A/S13A/S16A) with or without IKBKB. **(C)** Immunofluorescence showing decreased mutant HTT aggregation levels in the presence of IKBKB when S13/S16 residues are intact, with T3A modification further enhancing the capacity of IKBKB to reduce mutant HTT aggregation. Aggregate levels were assessed by GFP-signal acquisition and HTT levels by 2B7 antibody staining. (Representative images of n = 3 independent experiments. **(A)** For the HTT Ex1 Q72-EGFP WT ± IKBKB, the same images of (A) were reported. Scale bars represent 50 $\mu$m). **(C, D)** Quantitative analysis of images in (C). The fold change of each bar was calculated with respect to its own control without IKBKB. (Means and SDs were calculated on three biological replica. One-way analysis of variance, Dunnett's test [*$P$ < 0.05]).

studies in vivo remain to be demonstrated. As for investigational tools, several enzymes have been reported as modulators, and specifically IKBKB, TBK1, and CK2 for T3 and S13/S16 phosphorylation (Thompson et al, 2009; Atwal et al, 2011; Bustamante et al, 2015; Bowie et al, 2018; Ochaba et al, 2019; Hegde et al, 2020), PP1 for T3 phosphorylation (Branco-Santos et al, 2017), and AKT and SGK for S421 phosphorylation (Humbert et al, 2002; Rangone et al, 2004).

In general, however, these enzymes have not been demonstrated to potently, selectively, and quantitatively influence the relevant endogenous HTT PTM, largely due to the lack of suitable readouts capable of detecting a dynamic, labile protein modification which

likely affects only a portion of the total cellular protein pool. The availability and characterization of specific antibodies capable of sensitively detect HTT phosphorylation events (Aiken et al, 2009; Thompson et al, 2009; Atwal et al, 2011; Ansaloni et al, 2014; Cariulo et al, 2017), and their use in the recent development of novel ultrasensitive immunoassays (Cariulo et al, 2017, 2019) represents an appropriate platform for a more in depth validation of these current candidate modulators to provide deeper insights in the role of PTMs in HD. Of the candidate tools to increase N-terminal HTT phosphorylation, IKBKB represents one of the better validated enzymes reported to be involved in regulating pS13 or pS13/pS16 levels in cellular and animal models of HD (Thompson et al, 2009;

Atwal et al, 2011; Bustamante et al, 2015; Ochaba et al, 2019), including in an endogenous HTT context in mice (Ochaba et al, 2019). Canonical (IKK complex mediated) IKK signalling appears to be a contributor to HD toxicity mechanisms (Khoshnan & Patterson, 2011) and may control *HTT* transcription (Becanovic et al, 2015). Moreover, IKBKB inhibitors increase pS13 HTT levels and decrease toxicity in cellular models of HD (Atwal et al, 2011). Although these findings suggest that inhibition of canonical IKK signalling lead to decreased toxicity and lower expression of mutant HTT, conditional IKBKB knock-out worsens the phenotype in a mouse model of HD (Ochaba et al, 2019). These paradoxical findings suggest a degree of complexity which requires further investigation (Greiner & Yang, 2011), as indeed very little is known of the downstream signalling pathways leveraged by IKBKB to increase phosphorylation of the HTT N17 domain. IKBKB is a pleiotropic kinase capable of signalling via diverse pathways (Hacker & Karin, 2006; Perkins, 2007; Chau et al, 2008; Israel, 2010; Schrofelbauer et al, 2012) and may therefore contribute both positively and negatively to HD pathology, depending on cellular context and intracellular signalling homeostasis. Indeed, the illustration of a specific IKK pathway involved in the regulation of pS13 HTT levels adds an important aspect to the understanding of biological mechanisms involved in HD. Using a recently developed anti-pS13 antibody and ultrasensitive immunoassays (Cariulo et al, 2019), we have quantitatively examined the capacity of IKBKB to increase pS13 HTT levels expressed from the endogenous *HTT* locus in a human cell line widely used for HTT PTM studies. We thereby identified a mechanism regulating the activity of IKBKB on the pS13 HTT substrate and propose a molecular pathway whereby IKBKB phosphorylates pS13 HTT. Importantly, we found that the ability of IKBKB to modulate pS13 HTT levels was dependent on its catalytic activity as determined by both genetic (kinase domain inactivating mutation) and pharmacological (selective IKBKB inhibitor) means. As might be expected, the effects of IKBKB on pS13 HTT levels were regulated by phosphatases, such as PP2A, likely acting at the level of IKBKB rather than by dephosphorylating HTT S13 because IKBKB-induced but not basal pS13 HTT levels were unaffected by the presence of OA, PPP2CA RNAi, or PPP2CA overexpression. These observations are consistent with the described inhibitory effect of PP2A on IKBKB activity (DiDonato et al, 1997) and provide an additional level of complexity to data interpretation. Importantly, these findings offer a plausible explanation as to how pS13 HTT levels may be increased through inhibiting regulatory components of the kinase responsible for increasing pS13 HTT levels. To our knowledge, this is the first report which demonstrates a quantitative, catalytic activity-dependent, regulated modulation of endogenous HTT pS13 levels by IKBKB. The present work differentiates the IKBKB activity on pS13 HTT levels which we genetically demonstrate acts via an IKK complex independent pathway, from IKBKB/IKK involvement in mutant HTT toxicity and *HTT* transcription, believed to act via a canonical, IKK complex–mediated pathway (Khoshnan & Patterson, 2011; Becanovic et al, 2015). The characterization of the non-canonical, IRF-mediated IKK pathway as relevant for increasing pS13 HTT levels represents a significant advancement in the field and asks for further development of selective tools of non-canonical IKK pathway modulators to allow further research to elucidate mechanisms governing increases in pS13 HTT levels

without worsening mutant HTT toxicity (Khoshnan et al, 2004) or increase its expression (Becanovic et al, 2015). We further elucidated the mechanism by which IKBKB phosphorylates HTT, providing the first evidence that IKBKB interacts directly with HTT, and that this interaction is dependent on the catalytic activity of IKBKB but independent of IKBKB's HTT substrate phosphorylation. Finally, as a first piece of evidence that genetic modulation of pS13 HTT levels can reverse a pathologically relevant phenotype, we demonstrate, again for the first time, that increased pS13 HTT levels reduce mutant HTT aggregation in cells, fully phenocopying the effects of phosphor-mimetic mutations. This effect may be at least partly due to conformational changes in HTT protein (e.g., see [Cariulo et al, 2017; Daldin et al, 2017]) likely affecting the exposure of epitopes including within the polyQ and polyP regions and therefore influencing aggregation propensity.

In conclusion, we believe these findings provide a significant advance in the field in several ways. From a technical perspective, this is the first demonstration that it is possible to quantitatively measure and modulate endogenous levels of a pathologically relevant PTM in HTT, phenocopying the effects of phosphor-mimetic mutations on a disease relevant readout. Conceptually, our findings advance the understanding of the relevance of IKBKB in HD biology in several ways. First, through the identification of a regulatory, phosphatase-dependent mechanism acting via IKBKB, resulting in increased pS13 HTT levels. Second, through the identification of non-canonical, IRF-mediated IKK-signalling pathway, resulting in increased pS13 HTT levels distinct from the canonical, IKK complex–mediated pathway, whose activity contributes to HD and to regulation of HTT expression. These data provide a rationale for identifying selective, non-canonical IKK-signalling modulators and associated pharmacodynamic readouts, to further elucidate candidate pharmacological tools and pathways involved in HD pathophysiology.

# Materials and Methods

### Antibodies

The MW1, 2B7 and 4C9 antibodies bind to the polyQ stretch, the N17 domain and the polyproline region of HTT, respectively. They were obtained from the CHDI Foundation and their use in Western blotting and SMC assays has been described previously (Cariulo et al, 2017). The anti-pS13 antibody is an affinity purified, rabbit polyclonal antibody specific for the phosphorylated S13 residue of human huntingtin raised against the peptide LMKAFE(pS)LKSFQ developed by the CHDI Foundation and available from the Coriell Institute for Biomedical research, HD Community Biorepository (ID CH01115). The validation of this antibody in different applications has been formerly reported (Cariulo et al, 2019). The here used anti-pT3 antibody was previously described (Cariulo et al, 2017). Monoclonal antibodies mAb 2166 (catalog #MAB2166; Merck) and anti-GAPDH (catalog #G9545; Sigma-Aldrich), anti-FLAG mouse antibody (catalog #F1804; Sigma-Aldrich), anti-FLAG rabbit antibody (catalog #F7425; Sigma-Aldrich), anti-IKBKB antibody (catalog #ab32135; Abcam), anti-pS176/180-IKBKB antibody (catalog #2697;

Cell Signaling), anti-glial fibrillary acidic protein (catalog #G9269; Sigma-Aldrich), anti-PP2A antibody (catalog #SAB4200266; Sigma-Aldrich), anti-IRF3 antibody (catalog #AB76409; Abcam), and anti-pS386-IRF3 antibody (catalog #AB76493; Abcam) were all supplied by a commercial source. Secondary antibodies used for Western blotting were goat-anti-mouse IgG HRP conjugated (catalog #12-349; Merck) and goat-anti-rabbit IgG HRP conjugated (catalog #12-348; Merck). The D2-fluophore and terbium antibody labelling for TR-FRET assay were custom made at CisBio. The Alexa-647 labelling for detection used in SMC and in ELISA assays was performed using the Alexa Fluor-647 Monoclonal Antibody Labelling Kit from Thermo Fisher Scientific (catalog #A20186) following manufacturer's instructions. MW1 antibody was conjugated to magnetic particles for SMC assays, following the manufacturer's recommendations (catalog #03-0077-02; Merck).

## Plasmid and constructs

All plasmids used in this study have been ordered or custom synthetized at TEMA Ricerca or Genescript, respectively. The chosen vector for constructs IKBKB, IKBKB KD, IKBKB S/E, IKBKB NBD, IKBKB LZ is pCDNA3.1. IKBKA (#RC216718), IKBKG (#RC218044), IKBKE (#RC212481), and PPP2CA (#RC201334) are commercially sourced by TEMA Ricerca. These constructs are also FLAG-tagged at the C-terminus. cDNAs constructs encoding for N-terminal HTT fragments (exon 1 HTT 1-90, based on Q23 numbering; Table S1) bearing different polyQ lengths (Q16 or Q72) and mutations (T3A, S13A/S16A, T3D, S13D/S16D, T3A/S13A/S16A, T3D/S13D/S16D) codify also for a EGFP-Tag at the C-terminus and are inserted in a pCMV vector. N-terminal HTT fragment, HTT 1-571, based on Q23 numbering (Table S2), is encoded in an untagged pCDNA3.1 vector. Expression of all constructs in mammalian cells has been validated as previously reported (Fodale et al, 2014; Cariulo et al, 2019).

## HEK293T cell culture and manipulation

For Western blotting and immunoassay analysis, HEK293T cells were cultured, transfected, and lysed as previously described (Cariulo et al, 2019). For immunofluorescence analysis, HEK293T cells were plated in poly-L-lysine (100 $\mu$g/ml) coated 96-well black plates with $\mu$Clear bottom (catalog #655090; Greiner) at 20,000 cells per well and transfected using Lipofectamine 2000 (Thermo Fisher Scientific) according to manufacturer's protocols. Lysis was performed 24 h from transfection in lysis buffer (TBS, 0.4% Triton X-100) supplemented with 1X protease inhibitor cocktail (catalog #11697498001; Roche) and phosphatase inhibitor cocktail (catalog #04986837001; Roche). Okadaic acid (catalog #5934S; Cell Signaling) was delivered to HEK293T cells at a concentration of 350 nM for 1 h at 37°C before their lysis. The IKBKB inhibitor compound (Bay-65-1942 [Atwal et al, 2011]) was sourced by the CHDI Foundation and HEK293T cells were treated for 24 h at 37°C before lysis. The combination of the treatment with OA and Bay-65-1942 was performed as follows: HEK293T cells were transfected as previously reported; after 24 h from transfection, Bay-65-1942 compound was delivered to cells at the reported concentration, medium was changed and treatment was carried out for 24 h; 1 h before lysis, cells were treated with OA as previously described. Silencing via

RNAi was performed using specific PP2A-C$\alpha$ siRNA commercially supplied (catalog #sc-43509; Santa Cruz Biotechnology). The siRNA transfection was performed as reported above, using the siRNAs at the final concentration of 100 nM. Lysis of cells was carried out after 48 h from transfection.

## Western blot

Samples were denatured at 95°C in 4X loading Bbuffer (125 mM Tris–HCl, pH 6.8, 6% SDS, 4 M urea, 4 mM EDTA, 30% glycerol, 4% 2-mercaptoethanol and bromophenol blue) and loaded on NuPAGE 4–12% Bis–Tris Gel (catalog #WG1402BOX; Thermo Fisher Scientific). Proteins were transferred on PVDF membrane (catalog #162-0177; Bio-Rad Laboratories) using wet blotting. After fixing in 0.4% paraformaldehyde solution and blocking with 5% non-fat milk in TBS/0.1% Tween-20, primary antibodies incubation was carried out overnight at 4°C and secondary antibody incubations for 1 h at RT. Protein bands were detected using chemiluminescent substrate (Supersignal West Femto Maximum catalog #3406; Supersignal West Pico Maximum catalog #34087; Thermo Fisher Scientific) on Chemidoc XRS+ (Bio-Rad Laboratories). Densitometric analysis was performed using ImageJ software.

## Immunoprecipitation

Immunoprecipitation was performed using Dynabeads Protein G (catalog #10004D; Thermo Fisher Scientific) following the manufacturer's instructions and using anti-FLAG (mouse) antibody or an unrelated antibody (anti-glial fibrillary acidic protein) as previously described (Vezzoli et al, 2019). The pulled down material was loaded on a SDS–PAGE and the Western blot was performed as described above.

## SMC assay

50 $\mu$l/well of dilution buffer (6% BSA, 0.8% Triton X-100, 750 mM NaCl, and protease inhibitor cocktail) were added to a 96-well plate (catalog #P-96-450V-C; Axygen). Samples to be tested were diluted in artificial cerebral spinal fluid (ACSF: 0.3 M NaCl; 6 mM KCl; 2.8 mM CaCl$_2$-2H$_2$O; 1.6 mM MgCl$_2$-6H$_2$O; 1.6 mM Na$_2$HPO$_4$-7H$_2$O; 0.4 mM NaH$_2$PO$_4$-H$_2$O) supplemented with 1% Tween-20 and protease inhibitor cocktail, in a final volume of 150 $\mu$l/well. Finally, 100 $\mu$l/well of the MW1 antibody coupled with magnetic particles (appropriately diluted in Erenna assay buffer catalog #02-0474-00; Merck), were added to the assay plate, and incubated for 1 h at RT under orbital shaking for the capturing step. The beads were then washed with Erenna System buffer (catalog #02-0111-00; Merck) and resuspended using 20 $\mu$l/well of the specific detection antibody labeled with Alexa-647 fluorophore appropriately diluted in Erenna assay buffer (anti-pS13 antibody for pS13 HTT levels readout and 2B7 for total HTT levels readout). The plate was incubated for 1 h at RT under shaking. After washing, the beads were resuspended and transferred in a new 96-well plate. 10 $\mu$l/well of Erenna buffer B (catalog #02-0297-00; Merck) were added to the beads for elution and incubated for 5 min at RT under orbital shaking. The eluted complex was magnetically separated from the beads and transferred in a 384-well plate (Nunc catalog #264573; Sigma-Aldrich)

where it was neutralized with 10 $\mu$l/well of Erenna buffer D (catalog #02-0368-00; Merck). Finally, the 384-well plate was heat-sealed and analyzed with the Erenna Immunoassay System.

### TR–FRET assay

5 $\mu$l/well of samples and 1 $\mu$l/well of antibodies cocktail (2B7-Tb 1 ng/$\mu$l; FLAG-D2 10 ng/$\mu$l) diluted in lysis buffer (composition described before) were added to the 384-well assay plate (low volume-F Bottom catalog #784080; Greiner). Assay was performed according to (Fodale et al, 2014).

### ELISA assays

Assay plates (Nunc MaxiSorp 96-well polystyrene plate catalog #449824; Thermo Fisher Scientific) were coated with 100 $\mu$l/well of the capture antibody (2B7 or anti-PP2A) diluted in TBS at the final concentration of 1 $\mu$g/ml and incubated without shaking overnight at 4°C. After washing the plate three times with wash buffer (TBS, 0.1% Tween-20), the plate was blocked with 300 $\mu$l/well of BSA 1% in TBS (blocking buffer) for 30 min at RT and washed three times with wash buffer. 100 $\mu$l/well of the analyte diluted at the appropriate concentration in sample buffer (ACSF supplemented with 1% Tween-20 and protease inhibitor cocktail) was added to the wells and incubated for 1 h at RT. After three washing steps, 50 $\mu$l/well of detection antibody (anti-FLAG-D2) diluted in Erenna assay buffer (catalog #02-0474-00; Merck) at the final concentration of 1 $\mu$l/ml were added to the assay plate and incubated for 1 h at RT. After five washing steps, 20 $\mu$l/well of Erenna buffer B (catalog #02-0297-00; Merck) was added to the assay plate and incubated for 5 min at RT under orbital shaking. 20 $\mu$l/well of the neutralization Erenna buffer D (catalog #02-0368-00; Merck) were added to the assay plate. After 5 min of incubation at RT under mild agitation, 35 $\mu$l/well of the neutralized samples was transferred in a final 384-well plate (Nunc catalog #264573; Sigma-Aldrich). The 384-well plates were heat-sealed and analyzed with the Erenna Immunoassay System.

### High-content imaging and analysis

After 24 h from transfection (described before), medium was removed and HEK293T cells were washed three times with TBS and fixed for 20 min at RT in 4% paraformaldehyde/4% sucrose in TBS solution. After two washes in TBS, cells were incubated with blocking solution (1% donkey serum, 0.1% Triton-X, 3% BSA in TBS) for 1 h at RT and then incubated with primary 2B7 and anti-FLAG (rabbit) antibodies (previously described) diluted 1:300 in staining solution (0.2% Triton-X, 3% BSA in TBS) overnight at 4°C under constant mild agitation. The next day, cells were washed three times in TBS and incubated for 1 h at RT with donkey anti-mouse IgG antibody conjugated to Alexa-647 or donkey anti-rabbit IgG antibody conjugated to Alexa-555 (1:2,000 in staining solution) provided by a commercial source (catalog #A31571 and catalog #A31572; Life Technologies) together with 1 $\mu$g/ml of Hoechst staining solution (catalog #H33342; Sigma-Aldrich). After three washes with TBS, images were acquired in an automated fashion using the Incell-2000 high-content imaging system (GE-Healthcare) using a 20x objective and appropriate filters. Three independent experiments were performed. For each experiment, every condition was evaluated in technical duplicates, two well/conditions and five fields per well were acquired. Imaging time settings were: 50 ms for H33342 (DAPI), 30 ms for GFP, and 700 ms for Alexa-647 and Alexa-555 (2B7 and anti-FLAG). Images were analyzed using the Developer toolbox 1.9.1 software (GE-Healthcare). For counting of aggregates, a vesicle algorithm was used. Specifically, the vesicle segmentation settings panel is used to define the morphological parameters for the detection of vesicles, and to adjust the operation's sensitivity in the identification of these objects. Therefore, the aggregates were identified based on shape (form factor >0.8), size (Area—$\mu$m$^2$), and intensity of small objects.

### SMC data analysis and statistics

Normalized pS13 signal on total HTT levels measured by SMC assay was determined as the ratio between the fold increase obtained from MW1/pS13SMC signal (pS13 HTT levels readout) and the fold increase calculated on the MW1/2B7 SMC signal (total HTT levels readout). This ratio was derived as follows: MW1/pS13SMC assay and MW1/2B7 SMC assay were performed in parallel on the same samples which were analyzed in a serial dilution curve (six dilution points 1:3 plus blank, technical duplicates). For each readout, a curve fitting per sample (described by a four-parameter logistic curve fit) was calculated, and a fold increase among curves (which shared the top, the bottom, and the slope) was assessed basing on EC50 parameter (fixing one of the analyzed samples as a reference). The statistical significance was assessed using a paired $t$ test (two-tailed) in the case of the analysis of n = 2 samples, or using a one-way analysis of variance, applying Dunnett's test (which allows to compare all columns versus a fix control column) where n > 2 samples need to be compared. Graphs were generated and statistical analysis was performed using the software GraphPad Prism 6.

# Supplementary Information

# Acknowledgements

The work was funded by CHDI Foundation Inc., a non-profit biomedical research organization exclusively dedicated to collaboratively developing therapeutics that will substantially improve the lives of individuals with Huntington's disease.

## Author Contributions

C Cariulo: conceptualization, data curation, formal analysis, investigation, and writing—original draft.
P Martufi and M Verani: data curation, formal analysis, and investigation.
L Toledo-Sherman, R Lee, and C Dominguez: resources, supervision, and funding acquisition.

L Petricca and A Caricasole: conceptualization and writing—original draft and project administration.

## Conflict of Interest Statement

The authors declare that they have no conflict of interest.

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
