## [Reviewer comments · Life Science Alliance]

Life Science Alliance

IKBKB reduces huntingtin aggregation by phosphorylating Serine 13 via a non-canonical IKK pathway

Cristina Cariulo, Paola Martufi, Margherita Verani, Leticia Toledo, Ramee Lee, Celia Dominguez, Lara Petricca, and Andrea Caricasole

DOI: <https://doi.org/10.26508/lsa.202302006>

Corresponding author(s): Cristina Cariulo, IRBM S.p.A.

Review Timeline:

Submission Date:	2023-02-22
Editorial Decision:	2023-03-27
Revision Received:	2023-06-22
Editorial Decision:	2023-07-13
Revision Received:	2023-07-27
Accepted:	2023-07-31

Scientific Editor: Novella Guidi

Transaction Report:

March 27, 2023

Re: Life Science Alliance manuscript #LSA-2023-02006-T

Dr. Cristina Cariulo
IRBM S.p.A.
Via Pontina Km 30,600 - 00071 Pomezia (RM)
Pomezia 00071
Italy

Dear Dr. Cariulo,

Thank you for submitting your manuscript entitled "IKBKB reduces huntingtin aggregation by phosphorylating Serine 13 via a non-canonical IKK pathway" to Life Science Alliance. The manuscript was assessed by expert reviewers, whose comments are appended to this letter. We invite you to submit a revised manuscript addressing the Reviewer comments.

Thank you for this interesting contribution to Life Science Alliance. We are looking forward to receiving your revised manuscript.

Sincerely,

B. MANUSCRIPT ORGANIZATION AND FORMATTING:

Reviewer #1 (Comments to the Authors (Required)):

This manuscript is a very detailed examination of the modification of huntingtin 1-17 region, referred to as N17. They describe the modification of N17 at S13 by IKBKB, via a non-canonical IRF3 mediated pathway. In agreement with other data in the past by others, they see polyglutamine expansion leads to huntingtin hypo-phosphorylation, and that correcting this affects huntingtin fragment aggregation dynamics. They use some unique tools to focus on S13 modification alone, as well as T3, which has been understudied.

Minor points:

PolyQ is slang, and should be written as polyglutamine.

Introduction has some references as double brackets. i.e. (PTMs; (Ehrnhoefer, Sutton et al., 2011, Saudou & Humbert, 2016))

Major points, but there are too many to list.

Some of this introduction is outdated. It is not "widely believed to contribute significantly to HD through the propensity of such fragments to misfold and form aggregates". The reality is that the amyloid-like hypothesis of polyglutamine aggregation has been in question since 2006, where several groups have shown aggregation in HD fragment models to be protective. In more accurate models of HD, phenotypes are observed without any protein aggregation, and in clinical allele models (under 50 repeats), no aggregation is ever even observed, and no aggregation has been seen in iPSC-induced MSNs, despite clear phenotypes. Finally, human HD GWAS studies confirm over and over with massive statistical power that proteostasis pathways are not relevant to HD age of onset or severity, as most gene modifiers of this disease are involved in mitochondrial health of DNA damage repair. This is important because this is where current therapeutic efforts are focused in 2023.

As for exon1 short transcripts in HD. This still remains weak data in the hands of a single lab and not seen in much larger human transcriptomics efforts.

The manuscript confuses an 81 amino acid or 571 fragment model of huntingtin with huntingtin actually seen in human disease, a 3144aa protein. In 2023, we can no longer remove 82% of a disease protein and call it the same thing. It is reasons like this HD therapeutics research has suffered for 25 years, and currently, no compound has been developed to clinical trial phases worldwide based on fragment model systems. The test needs to more accurately refer to this as "HTT fragment model". This model also suffers from improper stoichiometry and typically relies on hyper alleles that are extremely rare in human disease, where even 55 CAG is an outlier in the population, or completely synthetic with pure CAG tracts also not typical of human alleles.

It was a model designed to produce aggregates, not model HD. Confusing fragment models with real HTT makes the manuscript difficult to follow.

Even in this manuscript, to study aggregation, they switch from the N571 to N81 to even see aggregates and they cannot see aggregates in HD cell lines with only endogenous HTT expressed because all of this is an artifact of overexpression and fragmentation, in a transformed an immortalized cell line with a proven history of poor reproduction because of the extremely high rate of mutation and genetic instability.

Intro and discussion needs to not gloss over the data from other that show Casein kinase 2 as a modifier of S13 and S16, which makes far more sense in HD pathology as CK2 is a known critical signaling pathway in DNA damage response, whereas again, nothing in GWAS implies a role for NFKB /IKB signaling at the level of disease age onset or severity.

They need to delineate N571 data from endogenous HTT with better labels in figures. It is good they looked at endogenous HTT, albeit in a problematic cell line.

There needs to be a caveat statement about signaling studies using over expressed kinases and substrates, because the data implies physiological stoichiometry is not important, when of course this isn't true. This methodology in kinase signaling is also outdated.

Figure 1 -

They are using HEK293T cells -these line has massively disrupted cell biology know relevant to HTT functions, including aberrant cell cycle, massive genotoxicity and instability. At least one experiment should be done on a human primary HD cell line, looking at the endogenous HTT. The biology of endogenous HTT in this line is extremely perturbed, which is why no one uses this line anymore in quality studies. It is unclear if the plasmids used in this study have an SV40 T origin of replication. If so, then they have a massive overexpression system. Huntingtin is not an abundant protein. The plasmids are poorly described in methods. At this point, these lines are primarily used as tools for protein over-expression for purification, not cell biology. This is one reason sources like Coriell and ATCC have developed non transformed lines for genetic disease studies.

All figures: the use of bar graph data representation is outdated. The explanation of this is in a J. Biol. Chem article but this is the reason many journals do no longer accept bar graphs as data representation.

DOI:<https://doi.org/10.1074/jbc.RA117.000147>

At the least , all data points should be included.

Figure 5: interaction is misspelled. I find this figure suspect, as kinase-substract interactions are extremely transient and it is unusual for a kinase to remain bound to substrate throughout the assays used, unless they are just aggregating two proteins, which is possible since no other control protein is co-expressed, and the overexpression concern above.

Overall, I found this manuscript extremely difficult to read, with large amounts of data that tell two stories, with some good data on actual huntingtin levels related to S13 modification, which has been seen by others with CK2, but certainly one substrate can have two signaling kinase modifiers. But a lot of aggregation data that in 2023 is proven irrelevant to this disease in a proven irrelevant model, which why is why these types of experiments tailed off in HD a decade ago.

I suggest they remove the Exon1 data and aggregation data, focus on HTT levels on endogenous huntingtin, and use primary human cells easily obtainable from Coriell for a few key experiments on HTT levels. They show that this signaling pathway may modify huntingtin, but they did not establish this modification is specific to IKBKB, and suffers from the same issues as previous studies where they only sought to establish this one pathway, by a variety of over-simplified tools and model systems. HTT level modulation is the primary therapeutic target in current HD clinical trials, so some of this data is important to publish, in a better context.

Reviewer #2 (Comments to the Authors (Required)):

Results from numerous independent studies in various experimental systems indicate that phosphorylation of residues T3/S13/S16 mitigates toxic effects of mutant huntingtin (mHTT) protein, suggesting that modulation of pathways promoting these phosphorylation events could represent a therapeutic strategy to alleviate Huntington's disease (HD) neuropathology. Unfortunately, specific kinases and phosphatases mediating homeostatic control of these residue's phosphorylation state have not yet been established, and potential cross-talk between individual phosphorylation sites have not yet been addressed. Here, Cariulo et al present work that sheds light on this important issue. This work involves a combination of transient protein expression, quantitative immunoassays to detect T3/S13/16 phosphorylation, as well as pharmacological and siRNA-based experiments using HEK293 cells as a model system. Significantly, results from these experiments reveals the IKK-related protein kinases IKBKB and IKBKE, but not the canonical IKK kinase IKBKA, as major regulators of S13 phosphorylation in these cells. Interestingly, these kinases were found to promote S13 phosphorylation indirectly through a mechanism involving modulation of the protein phosphatase PP2B, which in turn activates IKBKB and/or IKBKE by dephosphorylating activation-related residues in these kinases (S177/S181). Consistent with these findings, co-IP experiments confirmed an interaction of PP2B with IKBKB, and modulation of mHTT's S13 phosphorylation by this kinase was associated with activation of a non-canonical IKK pathway involving IRF3 activation. In addition, immunocytochemical experiments confirmed prior findings indicating that reduced S13 phosphorylation inversely correlates with mHTT aggregation.

Experiments in this manuscript are properly controlled and the data generated from these experiments deemed robust and sound. The paper is very nicely written, repeatedly acknowledging limitations of HEK cells as a cellular model to study HD pathogenesis, and clearly emphasizing the need of additional studies to evaluate findings in this model to more relevant HD-related experimental systems. In this reviewer's opinion, results from this work increase our understanding of mechanisms and specific molecular components underlying the regulation of HTT's amino-terminal phosphorylation (i.e., selected IKK kinases and PP2B), as well as the impact of these phosphorylation events on mHTT misfolding/aggregation. Overall, the findings bear potentially important pathophysiological implications for HD and should be of interest to Life Science Alliance readers. Authors should only address a few minor comments.

Minor comments.

Figs. 1A and 1E: The authors should speculate as to why does the anti-pS13 antibody recognizes full-length HTT doublet in this panel, and a single band in Fig. 1A.

Figs 2A and 2E: The authors should speculate as to why does IKBKB increase S13 phosphorylation to a greater extent in 2E than in 2A. Does the slight variability in the magnitude of effects result from transfection-related issues? (i.e., minor differences in the time of transfection prior to processing of samples?).

Discussion section: While the results presented unequivocally link the status of T3/S13/S16 phosphorylation to mHTT aggregation, the authors could briefly discuss a potential impact of these phosphorylation events on more discrete conformational changes (e.g., aberrant exposure of specific HTT exon 1 domains).

Referee Cross-Comments

I have carefully read all comments by Reviewer #1. He/she appears well informed of long-standing issues relevant to the HD field, and I agree with a few of his/her criticisms (see below). On the other hand, it is my belief our work as reviewers should be limited to judging rigor of the data and making sure that any conclusions derived from such data are consistent with the author's conclusions. With this notion in mind, I could not find comments from Rev#1 that would reveal any specific inconsistencies. A clear rationale supporting his/her comments is not provided, which makes those comments look more like personal views on HD pathogenesis, rather than constructive critiques. Further, the derogatory tone of some of his/her comments also struck me as biased.

Specific comments where I disagree with Rev#1.

The authors presented strong data documenting an impact of non-canonical IKK kinases/PP2B on T3/S13/S16 phosphorylation using pharmacological compounds, siRNA-based approaches, and well-controlled transient transfection experiments (i.e., IKK kinases with and without specific point mutations). The complementary use of those approaches strongly supports the author's main conclusions and represents a major strength of the manuscript.

Addressing a potential role of CK2 on T3/S13/S16 phosphorylation is well beyond the scope of this manuscript. Much of the published work addressing CK2's role on T3/S13/S16 phosphorylation would not have met Rev#1's standards.

The lack of GWAS studies implying a role of IKK kinases on HD pathogenesis does not invalidate any of the author's findings.

Removing all data from mHTT fragments, strikes me as an unfounded criticism and an implicit rejection of the manuscript. Most published reports using HEK cells to date have failed to consider endogenous full-length HTT, as the authors have done in this report. Further, most changes reported for mHTT fragments in this manuscript also extended to endogenous HTT.

Rev#2 refers to data in Figure 5 as "suspect", without providing any specifics, which strikes me as unfair. That specific point mutations in S13/S16 would completely abolish the interaction between overexpressed mHTT fragments and IKBKB strongly supports a (direct or indirect) protein-protein interaction. Based on these findings, future studies are likely to evaluate the extent to which this interaction extends to endogenous mHTT using more relevant HD models.

Specific comments where I agree with Rev#1.

I partially agree with Rev#1 on the limited relevance of HEK cells to study HD pathogenesis. On the other hand, a large body of evidence indeed supports the existence of mHTT fragments in various HD mouse models (see for example, work from the Bates and Hayden's labs), and much has been learned on the biology of mHTT fragments through their expression in different cell lines (including HEK cells). Importantly, limitations of HEK cells to understand HD pathogenesis are repeatedly acknowledged throughout the manuscript, and the authors go out of their way to emphasize the need of evaluating their findings in more relevant HD models, starting from the paper's Introduction section.

One specific point where I do agree with Rev#1 is that the quality of Figures could be improved by including all values.

Reviewer #3 (Comments to the Authors (Required)):

The manuscript by Cariulo et al. investigates the mechanism of S13 Huntingtin phosphorylation by IKBKB, its regulation, and its significance for Huntingtin-related cellular phenotypes. First, the authors demonstrate the involvement of IKBKB in phosphorylating S13 HTT and show that IKBKB activity is regulated by phosphatases such as PP2A. Second, they implicate the non-canonical IRF3-dependent IKK signaling pathway in this process. In addition, they uncover an interesting crosstalk between phosphorylation of several N-terminal residues, T3 and S13/S16. Importantly, they also show that these posttranslational modifications result in altered aggregation properties of mutant HTT in a cellular system and could therefore have therapeutic implications. The experiments are in general well designed, the methods and analyses are appropriate, the paper is clearly written, and the data is nicely presented in the figures.

Specific comments:

1. In Fig. 1C-D and E-F, the authors should also show a blot for IKBKB and pIKBKB to confirm its pharmacological inhibition. In addition, two bands are visible on the pS13 blot for endogenous Huntingtin in Fig. 1E-F - please explain the origin of the bands.
2. The claimed increase in endogenous pS13 HTT upon overexpression of IKBKB LZ that is mentioned in the text cannot be seen on the Western blots in Fig. 4A and C. The conclusion about monomeric IKBKB being able to phosphorylate S13, as well as the statement about complete correlation between pS13 HTT and pIRF3, are therefore not convincing. The authors should either demonstrate the effect in the figure or adjust the text to the presented data.
3. In Fig. 6A, as the aggregates are very big and bright, it is difficult to understand to which cells they belong. Please include overlay images of the three fluorescent channels, as well as overlaid zoomed-in higher-resolution images of some example aggregates.
Please also explain why the HTT aggregates are not recognized by the 2B7 antibody against the N-terminus of HTT?
4. In Fig. 6B, it is not clear how the analysis was performed, as the term "aggregate levels" is not defined, and the analysis is not sufficiently described in the Materials and Methods. What exactly was quantified here?
5. In Fig. 7A, please provide an explanation for the two bands visible for pS13 HTT in some of the lanes.
In Fig. 7C, statistical significance should be specified.
6. In Fig. 1 it is shown that overexpressed IKBKB only has an effect on the phosphorylation of overexpressed mutant HTT in the presence of OA treatment. In contrast, Fig. 7A-B and Fig. 8A-B show an effect of IKBKB on overexpressed HTT phosphorylation and aggregation in the absence of OA. How do the authors reconcile these observations?
7. The same control and IKBKB WT images were used in Fig. 8A and 8C - this should at least be acknowledged in the figure legend, as it seems that the data in panels A-B and C-D is not from independent experiments.
8. Fig. 8C is lacking an anti-HTT staining to demonstrate comparable levels of HTT expression in all the conditions.
9. Images in Figure S1 are of poor quality. Please replace with images of sufficient resolution.
10. While the main focus of the paper is on the phosphorylation of the S13 residue by IKBKB, the authors should at least discuss the potential contribution of other kinases to this phosphorylation event, as Fig. 1 demonstrates baseline phosphorylation of Huntingtin S13 even when IKBKB is inhibited.

Minor points:

1. The abstract contains several abbreviations (IKBKB, IKK, IRF3) that I think should be explained. I would also recommend explaining all the abbreviations when first mentioned in the main text.
2. First page of Introduction (by the way, page numbering and line numbering would be very helpful): the paper by Gu et al., Neuron 2015, is cited in the context of phospho-mimetic mutants, while it actually describes an N17-deletion mutant, so the citation would be more appropriate several lines earlier, in the sentence dealing with the importance of the N17 domain in cells and in vivo.
3. Second page of Introduction, second paragraph: The sentence "Studies using ... suggested that increased phosphorylation at T3 or S13/16 would need to be achieved" is not very clear, I recommend rephrasing, e.g. "would need to be achieved in order to decrease HTT toxicity".
4. Last paragraph of Introduction: "Coherently, ... IRF3 activation rather than through IKBA" - should be "IKBKA"?

Reviewer #1 (Comments to the Authors (Required)):

This manuscript is a very detailed examination of the modification of huntingtin 1-17 region, referred to as N17. They describe the modification of N17 at S13 by IKBKB, via a non-canonical IRF3 mediated pathway. In agreement with other data in the past by others, they see polyglutamine expansion leads to huntingtin hypo-phosphorylation, and that correcting this affects huntingtin fragment aggregation dynamics. They use some unique tools to focus on S13 modification alone, as well as T3, which has been understudied.

Minor points:

PolyQ is slang, and should be written as polyglutamine.

Introduction has some references as double brackets. i.e. (PTMs; (Ehrnhoefer, Sutton et al., 2011, Saudou & Humbert, 2016))

Major points, but there are too many to list.

Some of this introduction is outdated. It is not "widely believed to contribute significantly to HD through the propensity of such fragments to misfold and form aggregates". The reality is that the amyloid-like hypothesis of polyglutamine aggregation has been in question since 2006, where several groups have shown aggregation in HD fragment models to be protective. In more accurate models of HD, phenotypes are observed without any protein aggregation, and in clinical allele models (under 50 repeats), no aggregation is ever even observed, and no aggregation has been seen in iPSC-induced MSNs, despite clear phenotypes. Finally, human HD GWAS studies confirm over and over with massive statistical power that proteostasis pathways are not relevant to HD age of onset or severity, as most gene modifiers of this disease are involved in mitochondrial health of DNA damage repair. This is important because this is where current therapeutic efforts are focused in 2023.

As for exon1 short transcripts in HD. This still remains weak data in the hands of a single lab and not seen in much larger human transcriptomics efforts.

The manuscript confuses an 81 amino acid or 571 fragment model of huntingtin with huntingtin actually seen in human disease, a 3144aa protein. In 2023, we can no longer remove 82% of a disease protein and call it the same thing. It is reasons like this HD therapeutics research has suffered for 25 years, and currently, no compound has been developed to clinical trial phases worldwide based on fragment model systems. The test needs to more accurately refer to this as "HTT fragment model". This model also suffers from improper stoichiometry and typically relies on hyper alleles that are extremely rare in human disease, where even 55 CAG is an outlier in the population, or completely synthetic with pure CAG tracts also not typical of human alleles.

It was a model designed to produce aggregates, not model HD. Confusing fragment models with real HTT makes the manuscript difficult to follow.

Even in this manuscript, to study aggregation, they switch from the N571 to N81 to even see aggregates and they cannot see aggregates in HD cell lines with only endogenous HTT expressed because all of this is an artifact of overexpression and fragmentation, in a transformed and immortalized cell line with a proven history of poor reproduction because of the extremely high rate of mutation and genetic instability.

Intro and discussion needs to not gloss over the data from other that show Casein kinase2 as a modifier of S13 and S16, which makes far more sense in HD pathology as CK2 is a known critical

signaling pathway in DNA damage response, whereas again, nothing in GWAS implies a role for NFKB /IKB signaling at the level of disease age onset or severity.

They need to delineate N571 data from endogenous HTT with better labels in figures. It is good they looked at endogenous HTT, albeit in a problematic cell line.

There needs to be a caveat statement about signaling studies using over expressed kinases and substrates, because the data implies physiological stoichiometry is not important, when of course this isn't true. **This methodology in kinase signaling is also outdated.**

Figure 1 -

They are using HEK293T cells -these line has massively disrupted cell biology know relevant to HTT functions, including aberrant cell cycle, massive genotoxicity and instability. At least one experiment should be done on **a human primary HD cell line, looking at the endogenous** HTT. The biology of endogenous HTT in this line is extremely perturbed, which is why no one uses this line anymore in quality studies. It is unclear if the plasmids used in this study have an SV40 T origin of replication. If so, then they have a massive overexpression system. Huntingtin is not an abundant protein. The plasmids are poorly described in methods. **At this point, these lines are primarily used as tools for protein over-expression for purification, not cell biology.** This is one reason sources like Coreill and ATCC have developed non transformed lines for genetic disease studies.

All figures: the use of bar graph data representation is outdated. The explanation of this is in a J. Biol. Chem article but this is the reason many journals do no longer accept bar graphs as data representation.

DOI:<https://doi.org/10.1074/jbc.RA117.000147>

At the least, all data points should be included.

Figure 5: interaction is misspelled. I find this figure suspect, as kinase-substract interactions are extremely transient and it is unusual for a kinase to remain bound to substrate throughout the assays used, unless they are just aggregating two proteins, which is possible since no other control protein is co-expressed, and the overexpression concern above.

Overall, I found this manuscript extremely difficult to read, with large amounts of data that tell two stories, with some good data on actual huntingtin levels related to S13modification, which has been seen by others with CK2, but certainly one substrate can have two signaling kinase modifiers. But a **lot of aggregation data that in 2023 is proven irrelevant to this disease in a proven irrelevant model, which why is why these types of experiments tailed off in HD a decade ago.**

I suggest they remove the Exon1 data and aggregation data, focus on HTT levels on endogenous huntingtin, and use primary human cells easily obtainable from Coriell for a few key experiments on HTT levels. They show that this signaling pathway may modify huntingtin, but they did not establish this modification is specific to IKBKB, and suffers from the same issues as previous studies where they only sought to establish this one pathway, by a variety of over-simplified tools and model systems. HTT level modulation is the primary therapeutic target in current HD clinical trials, so some of this data is important to publish, in a better context.

1. This reviewer misses the point of the paper. We simply aim at characterizing the mechanism through which IKBKB, a kinase implicated in HD pathology of which HTT is a substrate *in vitro* and *in vivo*, phosphorylates HTT and produces consequences on a functional feature of mHTT (aggregation). To do this, we use a cellular model in order to dissect the different

signalling pathways downstream of IKBKB, an approach which would have been extremely difficult in the cellular systems proposed by this reviewer (whose real translational value has yet to be proven by a clinically validated drug, by the way). The demonstration of the relevance of the mechanisms described in this paper for HD pathology is outside the scope of this manuscript.

2. Further, this reviewer evidently does not subscribe to models of HD where N-terminal fragments of mutant HTT contribute to disease pathogenesis, in spite of numerous supporting reports and reviews (too many to mention here, but see e.g. some recent ones Kim et al., 2022 JCI Insight. 2022 Sep 8; 7(17): e154108.; Donnelly et al., 2022 Front Neurosci. 2022; 16: 946822 doi: 10.3389/fnins.2022.946822). There is nothing we can do about this, except point to the published evidence by more than “just one group” (as this reviewer states) and encourage the reviewer to be less dogmatic and more scientific in his approach. Some recent examples: A) Yang et al. 2020, <https://doi.org/10.1038/s41467-020-16318-1> where exon 1 HTT is shown to be stably present in the brain of HD140Q knock-in mice and leads to similar HD-like phenotypes and age-dependent HTT accumulation in the striatum and B) a series of papers demonstrating that the pathogenic exon 1 HTT protein is produced by incomplete splicing in Huntington's disease patients (Neuder et al., 2017 DOI: 10.1038/s41598-017-01510-z; Sathasivam et al., 2013 <https://doi.org/10.1073/pnas.1221891110>). In Neuder et al., is shown that a short HTT exon 1 mRNA comprised of Htt exon 1 and the 5' part of intron 1 is generated by incomplete splicing in HD knock-in mouse models and human HD brains. Irrespective of his/her opinion, it is a **fact** is that there is no mechanism validated by a clinically efficacious drug in HD as yet, and even GWAS-associated targets (if shown actually druggable) remain unproven to date. Only a clinically efficacious drug will validate the target mechanisms, and the models employed to progress them to the clinic, and until this happens there are only speculations.
3. Another aspect we encourage the reviewer to reconsider is his position on the use of cell lines in HD research. His criticism of the use of cell lines such as HEK293 to **model HD** is clearly relevant and agreeable by most HD scientists (including us) because cell lines are not useful to model human HD, but he is missing a key point. The use of cell lines and specifically of HEK293 cells (which express HTT endogenously) as a **cellular tool** (not as a disease model) is abundantly documented to study molecular mechanisms associated with mutant HTT production, post-translational modification, fragmentation, aggregation, clearance and even to identify candidate targets for modulating HTT levels (published literature to this effect is too large to mention here). In short, HEK293 cells have been amply deployed to study HTT biology and provide hypotheses to be tested in more evolved cellular models. Again, he misses the point of this manuscript (see 1 above).
4. His point on the formatting of the figures is well taken and we will review all figures.

Reviewer #2 (Comments to the Authors (Required)):

Results from numerous independent studies in various experimental systems indicate that phosphorylation of residues T3/S13/S16 mitigates toxic effects of mutant huntingtin(mHTT) protein, suggesting that modulation of pathways promoting these phosphorylation events could represent a therapeutic strategy to alleviate Huntington's disease (HD) neuropathology. Unfortunately, specific kinases and phosphatases mediating homeostatic control of these residue's phosphorylation state have not yet been established, and potential cross-talk between individual phosphorylation sites have not yet been addressed. Here, Cariulo et al present work that sheds light on this important issue. This work involves a combination of transient protein expression, quantitative immunoassays to detect T3/S13/16 phosphorylation, as well as pharmacological and siRNA-based experiments using HEK293 cells as a model system. Significantly, results from these experiments reveals the IKK-related protein kinases IKBKB and IKBKE, but not the canonical IKK kinase IKBKA, as major regulators of S13 phosphorylation in these cells. Interestingly, these kinases were found to promote S13 phosphorylation indirectly through a mechanism involving modulation of the protein phosphatase PP2B, which in turn activates IKBKB and/or IKBKE by dephosphorylating activation-related residues in these kinases (S177/S181). Consistent with these findings, co-IP experiments confirmed an interaction of PP2B with IKBKB, and modulation of mHTT's S13 phosphorylation by this kinase was associated with activation of a non-canonical IKK pathway involving IRF3 activation. In addition, immunocytochemical experiments confirmed prior findings indicating that reduced S13 phosphorylation inversely correlates with mHTT aggregation.

Experiments in this manuscript are properly controlled and the data generated from these experiments deemed robust and sound. The paper is very nicely written, repeatedly acknowledging limitations of HEK cells as a cellular model to study HD pathogenesis, and clearly emphasizing the need of additional studies to evaluate findings in this model to more relevant HD-related experimental systems. In this reviewer's opinion, results from this work increase our understanding of mechanisms and specific molecular components underlying the regulation of HTT's amino-terminal phosphorylation (i.e., selected IKK kinases and PP2B), as well as the impact of these phosphorylation events on mHTT misfolding/aggregation. Overall, the findings bear potentially important pathophysiological implications for HD and should be of interest to Life Science Alliance readers. Authors should only address a few minor comments.

We thank this reviewer for the appreciative comments and address his points individually below. He appears to be more knowledgeable in the field (and more objective in his review) than Reviewer 1. He also fully understands the contribution of the manuscript to the field as well as its limitations, which as he correctly stated we have amply discussed in the main text.

Minor comments.

Figs. 1A and 1E: The authors should speculate as to why does the anti-pS13 antibody recognizes full-length HTT doublet in this panel, and a single band in Fig. 1A.

Fig. 1 A represents an N-terminal fragment of HTT, while Fig. 3E represents FL HTT. In this latter case, anti-pS13 Ab detects a doublet, while only one band is detected by mAb2166. This is presumably due to differences in the nature of the detected HTT proteins, where differences in post-translational modifications, proteolytic pattern and protein stability can lead to variations in epitope recognition by different antibodies, as indeed described for some anti HTT antibodies (e.g. see Thomson et al., 2009 J Cell Biol. 2009 Dec 28;187(7):1083-99. doi: 10.1083/jcb.200909067).

Figs 2A and 2E: The authors should speculate as to why does IKBKB increase S13 phosphorylation to a greater extent in 2E than in 2A. Does the slight variability in the magnitude of effects result from transfection-related issues? (i.e., minor differences in the time of transfection prior to processing of samples?).

The differences on pS13 phosphorylation seen in Fig 2A vs Fig 2E can be ascribed to the different endogenous levels of phosphorylation of IKBKB which are variable as well as to the more “modest” increase of pS13 phosphorylation without the treatment with okadaic acid. Moreover, the figures should not be directly compared as in Fig 2A a scramble siRNA was used.

Discussion section: While the results presented unequivocally link the status of T3/S13/S16 phosphorylation to mHTT aggregation, the authors could briefly discuss a potential impact of these phosphorylation events on more discrete conformational changes (e.g., aberrant exposure of specific HTT exon 1 domains).

We have addressed this point in the discussion. Indeed, we and others have demonstrated that N17 phosphorylation can influence the structure of the N17 domain and the conformation and aggregation of HTT Exon 1 fragments (Cariulo et al., 2017 Proc Natl Acad Sci U S A. 2017 Dec 12;114(50):E10809-E10818. doi: 10.1073/pnas.1705372114; Daldin et al., 2017 Sci Rep. 2017 Jul 11;7(1):5070. doi: 10.1038/s41598-017-05336-7; Vieweg et al., 2021 J Mol Biol. 2021 Oct 15;433(21):167222. doi: 10.1016/j.jmb.2021.167222). Conformational changes associated with T3, S13/S16 phosphorylation may indeed change the availability of epitopes in the N17, polyQ and polyP region, for which relevant antibodies are available and which can consequently be investigated through different methodologies (immunoassays but also structural studies e.g. Cryo-EM as performed for MW1; e.g. Guo et al., 2018 Nature. 2018 Mar 1;555(7694):117-120. doi: 10.1038/nature25502). This is clearly outside the scope of the present manuscript but can clearly constitute an aspect for future investigations.

“Finally, as a first piece of evidence that genetic modulation of pS13 HTT levels can reverse a pathologically relevant phenotype, we demonstrate, again for the first time, that increased pS13 HTT levels reduce mutant HTT aggregation in cells, fully phenocopying the effects of phosphor-mimetic mutations. This effect may be at least partly due to conformational changes in HTT protein (e.g. see Cariulo et al., 2017; Daldin et al., 2017) likely affecting the exposure of epitopes including within the polyQ and polyP regions and therefore influencing aggregation propensity.”

Referee Cross-Comments

I have carefully read all comments by Reviewer #1. He/she appears well informed of long-standing issues relevant to the HD field, and I agree with a few of his/her criticisms (see below). On the other hand, it is my belief our work as reviewers should be limited to judging rigor of the data and making sure that any conclusions derived from such data are consistent with the author's conclusions. With this notion in mind, I could not find comments from Rev#1 that would reveal any specific inconsistencies. A clear rationale supporting his/her comments is not provided, which makes those comments look more like personal views on HD pathogenesis, rather than constructive critiques. Further, the derogatory tone of some of his/her comments also struck me as biased.

We thank Reviewer 2 and agree with his comments. Reviewer 1 does not seem to be objective but rather expression his/her poorly substantiated opinions.

Specific comments where I disagree with Rev#1.

The authors presented strong data documenting an impact of non-canonical IKK kinases/PP2B on T3/S13/S16 phosphorylation using pharmacological compounds, siRNA-based approaches, and well-controlled transient transfection experiments (i.e., IKK kinases with and without specific point mutations). The complementary use of those approaches strongly supports the author's main conclusions and represents a major strength of the manuscript.

Addressing a potential role of CK2 on T3/S13/S16 phosphorylation is well beyond the scope of this manuscript. Much of the published work addressing CK2's role on T3/S13/S16 phosphorylation would not have met Rev#1's standards.

The lack of GWAS studies implying a role of IKK kinases on HD pathogenesis does not invalidate any of the author's findings.

Removing all data from mHTT fragments, strikes me as an unfounded criticism and an implicit rejection of the manuscript. Most published reports using HEK cells to date have failed to consider endogenous full-length HTT, as the authors have done in this report. Further, most changes reported for mHTT fragments in this manuscript also extended to endogenous HTT.

Rev#2 refers to data in Figure 5 as "suspect", without providing any specifics, which strikes me as unfair. That specific point mutations in S13/S16 would completely abolish the interaction between overexpressed mHTT fragments and IKBKB strongly supports a (direct or indirect) protein-protein interaction. Based on these findings, future studies are likely to evaluate the extent to which this interaction extends to endogenous mHTT using more relevant HD models.

Specific comments where I agree with Rev#1.

I partially agree with Rev#1 on the limited relevance of HEK cells to study HD pathogenesis. On the other hand, a large body of evidence indeed supports the existence of mHTT fragments in various HD mouse models (see for example, work from the Bates and Hayden's labs), and much has been learned on the biology of mHTT fragments through their expression in different cell lines (including HEK cells). Importantly, limitations of HEK cells to understand HD pathogenesis are repeatedly acknowledged throughout the manuscript, and the authors go out of their way to emphasize the need of evaluating their findings in more relevant HD models, starting from the paper's Introduction section.

One specific point where I do agree with Rev#1 is that the quality of Figures could be improved by including all values.

Agreed. We have modified all figures accordingly.

Reviewer #3 (Comments to the Authors (Required)):

The manuscript by Cariulo et al. investigates the mechanism of S13 Huntingtin phosphorylation by IKBKB, its regulation, and its significance for Huntingtin-related cellular phenotypes. First, the authors demonstrate the involvement of IKBKB in phosphorylating S13 HTT and show that IKBKB activity is regulated by phosphatases such as PP2A. Second, they implicate the non-canonical IRF3-dependent IKK signaling pathway in this process. In addition, they uncover an interesting crosstalk between phosphorylation of several N-terminal residues, T3 and S13/S16. Importantly, they also show that these posttranslational modifications result in altered aggregation properties of mutant HTT in an acellular system and could therefore have therapeutic implications. The experiments are in general well designed, the methods and analyses are appropriate, the paper is clearly written, and the data is nicely presented in the figures.

Specific comments:

1. In Fig. 1C-D and E-F, the authors should also show a blot for IKBKB and pIKBKB to confirm its pharmacological inhibition.

We thank the reviewer for this suggestion and have now included this blot.

In addition, two bands are visible on the pS13 blot for endogenous Huntingtin in Fig. 1E-F- please explain the origin of the bands.

The origin of the two bands visible in pS13 WB might be due to a proteolytic cleavage that may specifically occur on full-length HTT protein endogenously expressed in HEK293 cells. The doublet is indeed not visible on overexpressed HTT-N571 fragment, that presumably might not be subjected to the same proteolytic process.

2. The claimed increase in endogenous pS13 HTT upon overexpression of IKBKB LZ that is mentioned in the text cannot be seen on the Western blots in Fig. 4A and C. The conclusion about monomeric IKBKB being able to phosphorylate S13, as well as the statement about complete correlation between pS13 HTT and pIRF3, are therefore not convincing. The authors should either demonstrate the effect in the figure or adjust the text to the presented data.

We thank the reviewer for pointing this out. We have now included a more representative WB.

3. In Fig. 6A, as the aggregates are very big and bright, it is difficult to understand to which cells they belong. Please include overlay images of the three fluorescent channels, as well as overlaid zoomed-in higher-resolution images of some example aggregates.

We have now provided also the merged figures, from where the absence of soluble huntingtin in the presence of aggregates is more visible.

Please also explain why the HTT aggregates are not recognized by the 2B7 antibody against the N-terminus of HTT?

One of the most widely accepted HTT aggregation model displays a compact architecture bearing a polyQ amyloid core, with the exposed PRD region sticking out and the covered N17 domain (Lin et al., 2017 Nat Commun. 2017 May 24;8:15462. doi: 10.1038/ncomms15462; Boatz et al., 2020 J Mol Biol. 2020 Jul 24;432(16):4722-4744. doi: 10.1016/j.jmb.2020.06.021). This model might explain the reason why 2B7 antibody is not able to recognize the aggregates, as its epitope may be masked by the aggregates structure.

4. In Fig. 6B, it is not clear how the analysis was performed, as the term "aggregate levels" is not defined, and the analysis is not sufficiently described in the Materials and Methods. What exactly was quantified here?

We have now provided the information on the applied analysis protocol in the M&M section.

5. In Fig. 7A, please provide an explanation for the two bands visible for pS13 HTT in some of the lanes.

The constructs employed for the experiments reported in Fig. 7A are all HTT GFP-tagged. The fusion with GFP might generate an IKBKB-dependent phosphor-site, which is recognized by pS13 antibody. The doublet is indeed detectable only in samples containing the overexpressed IKBKB.

In Fig. 7C, statistical significance should be specified.

We have now included significance in the figures.

6. In Fig. 1 it is shown that overexpressed IKBKB only has an effect on the phosphorylation of overexpressed mutant HTT in the presence of OA treatment. In contrast, Fig. 7A-B and Fig. 8A-B show an effect of IKBKB on overexpressed HTT phosphorylation and aggregation in the absence of OA. How do the authors reconcile these observations?

The HTT constructs used in the experiments for Fig. 1 (HTT-N571) are different from those employed in Fig 7 and 8 (HTT-Ex1), thus the figures should not be directly compared since a different degree of phosphorylation on Ex1 with respect to N571 fragment might not be excluded.

7. The same control and IKBKB WT images were used in Fig. 8A and 8C - this should at least be acknowledged in the figure legend, as it seems that the data in panels A-B and C-D is not from independent experiments.

We have now acknowledged that in the figure's text.

8. Fig. 8C is lacking an anti-HTT staining to demonstrate comparable levels of HTT expression in all the conditions.

We have now included the relevant pictures.

9. Images in Figure S1 are of poor quality. Please replace with images of sufficient resolution.

We have now increased the quality of the pictures.

10. While the main focus of the paper is on the phosphorylation of the S13 residue by IKBKB, the authors should at least discuss the potential contribution of other kinases to this phosphorylation event, as Fig. 1 demonstrates baseline phosphorylation of Huntingtin S13 even when IKBKB is inhibited.

Agreed. We have now included a sentence in the discussion to this effect including reference to all HTT candidate kinases reported thus far in the literature.

"As for investigational tools, several enzymes have been reported as modulators, and specifically IKBKB, TBK1 and CK2 for T3 and S13/S16 phosphorylation (Atwal et al., 2011, Bowie et al., 2018, Bustamante et al., 2015, Hegde et al., 2020, Ochaba et al., 2019, Thompson et al., 2009), PP1 for T3 phosphorylation (Branco-Santos et al., 2017), and AKT and SGK for S421 phosphorylation (Humbert, Bryson et al., 2002, Rangone, Poizat et al., 2004)."

Minor points:

1. The abstract contains several abbreviations (IKBKB, IKK, IRF3) that I think should be explained. I would also recommend explaining all the abbreviations when first mentioned in the main text.

For brevity reasons we have made the suggested changes in the introduction section.

2. First page of Introduction (by the way, page numbering and line numbering would be very helpful): the paper by Gu et al., Neuron 2015, is cited in the context of phospho-mimetic mutants, while it actually describes an N17-deletion mutant, so the citation would be more appropriate several lines earlier, in the sentence dealing with the importance of the N17 domain in cells and in vivo.

Agreed.

3. Second page of Introduction, second paragraph: The sentence "Studies using ...suggested that increased phosphorylation at T3 or S13/16 would need to be achieved" is not very clear, I recommend rephrasing, e.g. "would need to be achieved in order to decrease HTT toxicity".

Agreed.

4. Last paragraph of Introduction: "Coherently, ... IRF3 activation rather than through IKBA" - should be "IKBKA"?

I κ B α (nuclear factor of kappa light polypeptide gene enhancer in B-cells inhibitor, alpha). Now we explained the abbreviation in the main text.

July 13, 2023

RE: Life Science Alliance Manuscript #LSA-2023-02006-TR

Dr. Cristina Cariulo
IRBM S.p.A.
Via Pontina Km 30,600 - 00071 Pomezia (RM)
Pomezia, Pomezia 00071
Italy

Dear Dr. Cariulo,

Thank you for submitting your revised manuscript entitled "IKBKB reduces huntingtin aggregation by phosphorylating Serine 13 via a non-canonical IKK pathway". We would be happy to publish your paper in Life Science Alliance pending final revisions necessary to meet our formatting guidelines.

- please correct the typo pointed out by Reviewer 3
- please add ORCID ID for the corresponding (and secondary corresponding) author--you should have received instructions on how to do so
- please add the Twitter handle of your host institute/organization as well as your own or/and one of the authors in our system
- please correct the name discrepancy for one of your co-authors (Leticia Toledo-Sherman in ms. file vs. Leticia Sherman Toledo in the system)
- please use the [10 author names et al.] format in your references (i.e., limit the author names to the first 10)
- please upload your Supplementary Tables in editable .doc or Excel format
- Tables should be numbered consecutively with Arabic numerals (S1, S2), and please correct their callouts in the manuscript text accordingly

A. FINAL FILES:

B. MANUSCRIPT ORGANIZATION AND FORMATTING:

Sincerely,

Reviewer #2 (Comments to the Authors (Required)):

The authors have provided a strong response to mine and the other reviewer's critiques, and modified the paper accordingly. In my opinion, the manuscript should now be ready for publication.

Reviewer #3 (Comments to the Authors (Required)):

In the revised version of the paper by Cariulo et al., the authors have thoroughly addressed the specific comments of all the reviewers, and made the requested modifications in the manuscript. The figures have improved in quality, and also gained in data transparency due to the inclusion of the single data points.

In my opinion, the paper looks great, all the conclusions are clear and supported by the data, and the manuscript deserves to be published in its current form.

A minor point: There is a typo in Fig. 5B, "interacion" should be "interaction".

July 31, 2023

RE: Life Science Alliance Manuscript #LSA-2023-02006-TRR

Dr. Cristina Cariulo
IRBM S.p.A.
Via Pontina Km 30,600 - 00071 Pomezia (RM)
Pomezia, Pomezia 00071
Italy

Dear Dr. Cariulo,

Thank you for submitting your Research Article entitled "IKBKB reduces huntingtin aggregation by phosphorylating Serine 13 via a non-canonical IKK pathway". It is a pleasure to let you know that your manuscript is now accepted for publication in Life Science Alliance. Congratulations on this interesting work.

DISTRIBUTION OF MATERIALS:

Again, congratulations on a very nice paper. I hope you found the review process to be constructive and are pleased with how the manuscript was handled editorially. We look forward to future exciting submissions from your lab.

Sincerely,
